



# Diurnal evolution of negative atmospheric ions above the boreal forest: From ground level to the free troposphere

Lisa J. Beck[1], Siegfried Schobesberger[2], Heikki Junninen[1,3], Janne Lampilahti[1], Antti Manninen[4], Lubna Dada[1,5,6], Katri Leino[1], Xu-Cheng He[1], Iida Pullinen[2], Lauriane Quéléver[1], Anna Franck[1], Pyry Poutanen[1], Daniela Wimmer[1], Frans Korhonen[1], Mikko Sipilä[1], Mikael Ehn[1], Douglas R. Worsnop[1,7], Veli-Matti Kerminen[1], Tuukka Petäjä[1,9], Markku Kulmala[1,8,9,10], and Jonathan Duplissy[1,11]

[1]Institute for Atmospheric and Earth System Research/Physics, Faculty of Science, University of Helsinki, 00014 Helsinki, Finland
[2]Department of Applied Physics, University of Eastern Finland, 70211 Kuopio, Finland
[3]Institute of Physics, University of Tartu, Tartu, 50090, Estonia
[4]Finnish Meteorological Institute, Helsinki, Finland
[5]Extreme Environments Research Laboratory, Ecole Polytechnique Fédérale de Lausanne (EPFL) Valais, Sion, 1951, Switzerland
[6]Laboratory of Atmospheric Chemistry, Paul Scherrer Institute, Villigen, 5232, Switzerland
[7]Aerodyne Research Inc., Billerica, MA, USA
[8]Aerosol and Haze Laboratory, Beijing Advanced Innovation Center for Soft Matter Sciences and Engineering, Beijing University of Chemical Technology (BUCT), Beijing, China
[9]Joint International Research Laboratory of Atmospheric and Earth System Sciences, School of Atmospheric Sciences, Nanjing University, Nanjing, China
[10]Faculty of Geography, Lomonosov Moscow State University, Moscow, Russia
[11]Helsinki Institute of Physics (HIP) / Physics, Faculty of Science, University of Helsinki, 00014 Helsinki, Finland

**Correspondence:** Lisa Beck (lisa.beck@helsinki.fi), Jonathan Duplissy (jonathan.duplissy@helsinki.fi), and Tuukka Petäjä (Tuukka.petaja@helsinki.fi)

**Abstract.** At SMEAR II research station in Hyytiälä, located in the Finnish boreal forest, the process of new particle formation and the role of ions has been investigated for almost 20 years near the ground and at canopy level. However, above SMEAR II, the vertical distribution and diurnal variation of these different atmospheric ions are poorly characterized. In this study, we assess the atmospheric ion composition in the stable boundary layer, residual layer, mixing layer and free troposphere, and the

evolution of these atmospheric ions due to photochemistry and turbulent mixing through the day. To measure the vertical profile of atmospheric ions, we developed a tailored setup for online mass spectrometric measurements, capable of being deployed in a Cessna 172 with minimal modifications. Simultaneously, instruments dedicated to aerosol properties measured in a second Cessna. We conducted a total of 16 measurement flights in May 2017, during the spring, which is the most active new particle formation season. A flight day typically consisted of three distinct flights through the day (dawn, morning and afternoon) to

observe the diurnal variation and at different altitudes (from 100 m to 3200 m above ground), and to capture the boundary layer development from stable boundary layer, residual layer to mixing layer, and the free troposphere. Our observations showed that the ion composition is distinctly different in each layer and depends on the air mass origin and time of the day. Before sunrise, the layers are separated from each other and have their own ion chemistry. We observed that the ions present within





the stable layer are of the same composition as the ions measured at the canopy level. During daytime when the mixing layer

evolved and the compounds are vertically mixed, we observed that highly oxidised organic molecules are distributed to the top of the boundary layer. The ion composition in the residual layer varies with each day, showing similarities with either the stable boundary layer or the free troposphere. Finally, within the free troposphere, we detected a variety of carboxylic acids and ions that are likely containing halogens, originating from the Arctic Sea.

## 1 Introduction

Atmospheric ions have been a subject of research for many decades (Hõrrak et al., 1994; Tammet, 1995; Hirsikko et al., 2011, and references therein). The term 'atmospheric ion' includes molecular ions and ion clusters. Ion clusters are composed of a neutral molecule or molecular cluster attached to a molecular ion. The source of ions varies. In higher altitudes, ions are primarily produced via galactic cosmic ray interaction with the atmosphere, while closer to the ground, radon decay and gamma radiation from soil are the main sources of ions (Israël, 1970). The primarily formed ions, such as $N_2^+$, $O_2^+$ and $e^-$, quickly

undergo reactions to form secondary molecular ions (Arnold, 2006; Junninen et al., 2016). Depending on the proton affinity of the trace gases present in the atmosphere, different compounds can be charged, leading to a variety of ions with different elemental compositions (Mohnen, 1976; Hirsikko et al., 2011; Ehn et al., 2010). The main sinks of atmospheric ions are ion-ion recombination and attachment to pre-existing aerosol particles (Israël, 1970; Smith and Spanel, 1995; Arnold, 2006).

The role of ions in the atmosphere is broad. For example, ions can alter cloud microphysics through enhancing condensation or coalescence with rain droplets (Harrison and Ambaum, 2009). Ions can also amplify the formation of secondary aerosols. Various field studies have shown that, depending on the location, ion-induced nucleation initiates about 1 to 100 % of new particle formation (NPF) (Hirsikko et al., 2011; Manninen et al., 2010; Jokinen et al., 2018; Beck et al., 2021a). Atmospheric NPF produces secondary aerosol particles via gas-to-particle conversion, including clustering and further growth of these clus-

ters to larger sizes. Ions enhance NPF, mainly by acting as stabilizers for clusters, by reducing the critical cluster size, and facilitating their growth to larger sizes (Seinfeld and Pandis, 2016; Kirkby et al., 2016; Wagner et al., 2017; Rose et al., 2018; Merikanto et al., 2016). Secondary aerosol particles affect human health (Baltensperger et al., 2008; Lelieveld et al., 2015), interact directly with radiation and act as cloud condensation nuclei (CCN) or ice nuclei (IN) (IPCC, 2021; Merikanto et al., 2009; Seinfeld and Pandis, 2016). In terms of their number concentration, NPF is the dominant source of aerosol particles

in the global atmosphere (e.g. Gordon et al., 2017), but both concentration and chemical composition of secondary aerosol particles vary considerably with location (Zhang et al., 2007). Globally, it is estimated that roughly half of the CCN is derived from NPF (Merikanto et al., 2009; Gordon et al., 2017). The importance of secondary aerosol particle in aerosol-radiation and aerosol-cloud interactions necessitates a thorough understanding of their formation from precursor gases (Lee et al., 2019).

Ion-induced nucleation mechanisms have been studied theoretically (Laakso et al., 2002), experimentally and via field measurements. Laboratory studies, such as the CLOUD experiment (Kirkby et al., 2011; Duplissy et al., 2016), investigated the





molecular processes leading to NPF and the role of ions within these processes. These experimental data have revealed different molecular pathways and quantified the importance of charged and neutral processes in the different molecular formation mechanisms. Such mechanisms include NPF associated with sulfuric acid (Duplissy et al., 2016; Merikanto et al., 2016), sulfuric

acid and ammonia (Kirkby et al., 2011; Kürten, 2019), sulfuric acid and amines (Almeida et al., 2013), iodine oxoacids (He et al., 2021), various organic compounds (Riccobono et al., 2014; Kirkby et al., 2016), and mixtures of organic compounds, ammonia, and sulfuric acid (Lehtipalo et al., 2018). The two latter mechanisms involve the formation of highly oxygenated organic molecules (HOM), in particular via the autoxidation of volatile organic compounds (VOC) (Ehn et al., 2014; Bianchi et al., 2019). The lower volatility of the HOM compared to the VOC enables them to contribute also to particle growth through

condensing onto nucleated particles (Kulmala et al., 1998; Ehn et al., 2014; Bianchi et al., 2019; Donahue et al., 2012). While HOM are a significant contributor to NPF in the boreal forest environment (Yan et al., 2018), their contribution to the particle growth was also observed in the Arctic (Beck et al., 2021a).

Ions have been observed to influence NPF in several environments. In coastal Antarctica, negative ion-induced nucleation

of sulfuric acid and ammonia was identified as the main NPF mechanism (Jokinen et al., 2018). In the Arctic region, several NPF pathways have been observed (Schmale and Baccarini, 2021), many of them involving ions. These include sulfuric acid-ammonia nucleation and iodic acid nucleation (Beck et al., 2021a; Baccarini et al., 2020), with subsequent particle growth being also influenced by condensation of methanesulfonic acid (MSA) or HOM. In the boreal forest environment, ion-induced nucleation was first reported by Kulmala et al. (2004), and the daytime ion-induced nucleation pathway was found to involve

sulfuric acid, ammonia and HOM (Yan et al., 2018). During evening hours when sulfuric acid concentrations are rather low, particle formation up to 6 nm by clustering of HOM was observed in the boreal forest (Rose et al., 2018).

Further up in the atmosphere, field studies conducted at mountain stations have provided insight into the behaviour of ions in the transition between the boundary layer and the free troposphere. Rose et al. (2015) observed that NPF at Puy de Dôme

(1465 m asl), France, was occurring in the interface of the boundary layer and free troposphere. However, the first steps of NPF appeared to be driven by neutral clusters rather than ions. Bianchi et al. (2016) reported the composition of negative ions during NPF days at Jungfraujoch (3580 m asl), Switzerland. They observed organic compounds in the free troposphere when the measured air mass had been in contact with the boundary layer before reaching the measurement site. A further study by Bianchi et al. (2021) observed a high frequency of NPF in the Himalayas at the Nepal Climate Observatory Pyramid station

(5079 m asl), where up-valley winds carry the precursor gases upwards, delivering them to the free troposphere and enabling nucleation. An airborne study (ATom) by Williamson et al. (2019) observed NPF at high altitudes above the tropics, initiated in the upper troposphere. However the chemical composition of the initial clusters remained unknown.

Atmospheric processes in the boundary layer, such as roll vortices, are a relevant vertical transport mechanism and carry

vapours and aerosols from lower levels to higher altitudes and vice versa (Etling and Brown, 1993). Using airborne measurements above a boreal forest, Lampilahti et al. (2020) showed that roll vortices can enhance NPF and lead to the particles'





growth up to CCN sizes. However, the compositions during such NPF events were not elucidated. Atmospheric chemistry and secondary aerosol formation are taking place throughout the troposphere, so continuous mixing and both vertical and horizontal transport of air masses should be considered when conducting field studies. The majority of detailed free-tropospheric aerosol observations have been made in mountain stations, which are still somehow influenced by transportation from the boundary layer.


Our present study combines 1) airborne aerosol and ion cluster measurements, 2) ground-based in situ observations and 3) remote sensing observations. Our goals are firstly to map the differences of atmospheric ions between the stable boundary layer, residual layer, mixing layer and free troposphere, and secondly, to explore how the character of atmospheric ions changes with photochemistry and turbulent mixing condition during daytime, and how this influences the vertical distribution of ions within the boundary layer, and thirdly to probe for possible ions precursors of NPF within the boundary layer and free troposphere.


## 2 Methods

The presented data were collected during an airborne measurement campaign in spring 2017 on six intensive measurement days between 2 and 16 May above the boreal forest at the Station for Measuring Ecosystem Atmospheric Relations (SMEAR) II, located in Hyytiälä, southern Finland (Hari and Kulmala, 2005; Brasseur et al., 2021). In addition, continuous ground-based data from SMEAR II were included in this study, such as aerosol precursor gases and ions, particle size distributions, meteorological data and Doppler lidar (section 2.4). The days of airborne measurements were chosen with the help of the ground measurements from SMEAR II, using the NPF prediction from Nieminen et al. (2015).



The airborne measurements were conducted with two Cessna 172, one equipped with a mass spectrometer measuring in negative ion mode (section 2.1) and the second aircraft equipped with particle and meteorological measurement devices (section 2.2). Both aircraft were flying at the same time and in formation above the boreal forest, to simultaneously capture vertical profiles of ion composition and particle size distribution.

### 2.1 Airborne measurements with APi-TOF


We deployed an Atmospheric Pressure Interface Time-of-Flight mass spectrometer (APi-TOF, Tofwerk AG  Aerodyne Research Inc.; see Junninen et al., 2010) in a high-performance Cessna 172 aircraft (Reims FR172) to detect negative ions and ion clusters within the boundary layer and the free troposphere. The principal components of an APi-TOF are an atmospheric pressure interface, attached to a time-of-flight (TOF) chamber. The atmospheric pressure interface allows the transition of ions from ambient pressure to the high vacuum in the TOF. The atmospheric pressure interface consists of three chambers with various ion optics and pressures. The sample is transported from ambient pressure to the first chamber (Small Segmented Quadrupole, SSQ) at ~2 mbar pressure (the pressure of the SSQ was controlled during the flight measurements, see end of this section), then further to the Big Segmented Quadrupole (BSQ) at ~$10^{-3}$ mbar, the Primary Beam (PB) chamber at ~$10^{-4}$




mbar and finally to the TOF at ~$10^{-6}$ mbar. Inside the TOF, the ions are guided to a detector, acquiring their time of flight

from which the mass per charge (m/z) is calculated (Junninen et al., 2010). The total ion sample rate to the instrument was set

to 20 lpm (more details at the end of this section) of which 0.8 lpm is entering the APi-TOF through a pinhole with a diameter

of 0.3 mm.

A tailored setup for the APi-TOF, including its own power supply, had to be developed to conduct the measurements on-

board the Cessna aircraft and to maintain the vacuum inside the APi-TOF between the flights while, for example, the Cessna

was parked in the airport hangar or for refuelling. This standalone independent setup was particularly designed to offer the

possibility to deploy the mass spectrometer in most single-engine Cessna models, which are widely used around the globe, and

to allow further measurements in different parts of the world.

The maximum weight for the tailored instrumentation setup that can be installed in the above-mentioned Cessna, while

allowing for the required fuel and a crew of two, was 200 kg. The APi-TOF (85 kg) was first mounted in a rack (Fig. 1c),

using helical wire rope isolators to prevent the vibration of the airplane from affecting the mass spectrometer components.

The specifically designed outer aluminium rack (12.7 kg) was constructed to altogether hold the APi-TOF, the power supply

and the EBARA Dry Vacuum Pump (PDV500, EBARA Corporation, 20 kg) used for the pre-vacuum in the APi-TOF. The

whole construction can be fit on the seat rails in the back of any Cessna 172 (or larger models) from which the back seats are

removed. The power for all the scientific instrumentation, including the mass spectrometer, computer, vacuum pump and addi-

tional measurement devices, was provided by a lithium iron phosphate battery (MLI Ultra 24/5500, Mastervolt, 57 kg and 23

kg for additional battery components). The lithium iron-phosphate battery was chosen as the safest solution for this campaign,

which required a high energy density. This type of battery can withstand temperatures of up to 270°C before catching fire,

whereas a common lithium cobalt oxide battery already overheats at 150°C. Furthermore, it has been tested in a set of standard

stress tests and fulfils the test requirements according to UN38.3. An automatic fail-safe switch was implemented to shut down

the device in case of any overheating or battery emergency. In addition, a manual safety switch was mounted in the cockpit, so

that the pilot or instrument operator could shut down the battery conveniently.

The battery lifetime was sufficient to cover the 2.5-hour flight duration, the refuelling of the Cessna, and additional possible

delays leading to prolonged flight times. For safety reasons, the battery shutdown threshold was set to 20%; the battery status

reached on average 40% after a typical flight of 2.5–3 hours (see Fig. S1, supplementary material). In between flights, the

batteries were recharged in the hangar directly inside the Cessna within two hours.

During both take-off and landing, the inlet system had to be located inside the airplane. Therefore, a sliding inlet-system of

110 cm in length and 5 kg in weight (Fig. 1a, c) was mounted on the top of the rack. The operator could then connect the inlet

of the APi-TOF with the inlet-line during the flight. At the entrance of the inlet line we used a stainless-steel T-shaped inlet





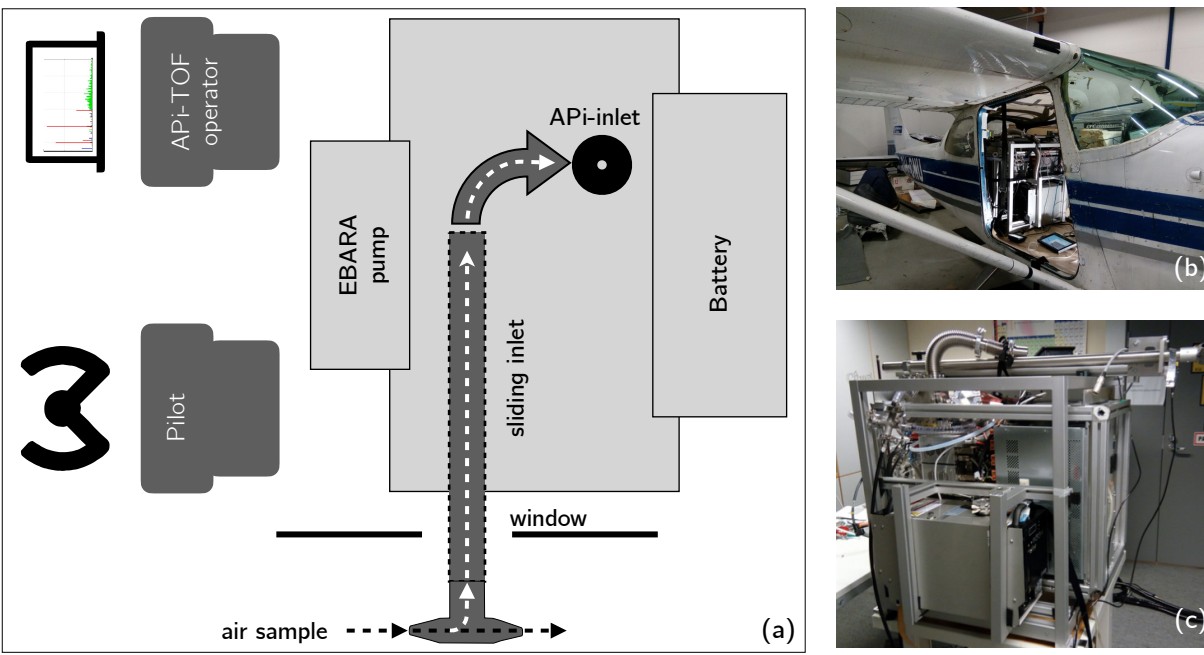

**Figure 1.** Setup of the APi-TOF inside the Cessna 172 aircraft. (a) Schematic of the APi-TOF setup from top. The air sample was taken through a T-piece to reduce turbulence and guide the sample into the APi-TOF pinhole through a sliding inlet system. The battery was in the back of the airplane; the vacuum pump (EBARA) was located behind the pilot and co-pilot seat. (b) Cessna 172 with the APi-TOF setup in the rear. (c) APi-TOF setup with outer rack, holding the pump and battery and the sliding inlet construction.

(Mirme et al., 2010) to reduce turbulence and to guide the air sample smoothly into the sliding inlet system (Fig. 1a).

Due to restrictions regarding chemicals onboard the aircraft, we could not use a chemical ionisation method to detect neutral molecules. Instead, we measured with a direct Ambient Ionisation (AI) inlet combined with an APi-TOF (AI-APi-TOF, Sahyoun et al., 2019), using a soft x-ray source (Hamamatsu, L12535) to ionise the air sample before entering the mass spectrometer. The x-ray source was operated for 20 seconds within a 10-minute interval. The acquired data with the x-ray ionisation were primarily used to determine the peak shape and to perform an instrumentation check during the flight measurements. The

signal with ionisation gave ~$10^6$ ions per second, instead of only ~10 ions per second without ionisation. This allowed the operator to evaluate if the ion count was low due to natural reasons, or if the instrumentation was malfunctioning.

     At higher altitudes, the dropping ambient pressure affected the pressures inside the chambers of the APi-TOF. Therefore, an IMR pressure controller (Aerodyne Research Inc.) was attached to the SSQ chamber to maintain pressure at 2 mbar by

introducing compensatory air into the chamber (Fig. S2). However, the lower pressures in the subsequent instrument chambers





(BSQ, PB and TOF) decreased along with decreasing ambient pressure by ~20% (BSQ) and ~40% (TOF) even with stable SSQ pressure. As the IMR controller malfunctioned on the first two measurement days (2 and 3 May 2017, Fig. S2), we were able to compare if the data were affected by stable SSQ or decreasing SSQ pressure, respectively. In both cases (stable or not stable pressure withing SSQ), the total ion count at the different height levels did not seem to be affected. However, for quantitative
measurements, further investigation in a controlled lab environment would be necessary in order to understand the effect of pressure on the amount of signal.

The total ion sample flow was kept at 20 lpm. The function of the different pumps attached to an APi-TOF were slightly adapted: The 20 lpm ion sample flow was created by the MD1 (Fore) pump of the APi-TOF setup, which originally was used
to support the Turbo pump. Instead, the support for the Turbo pump was provided by the external EBARA vacuum pump. This configuration saved additional weight, as no extra pump for the inlet flow was necessary.

The aircraft was also equipped with a GPS receiver and a temperature, relative humidity and pressure sensor for meteorological data analysis. The overall weight of the setup including all components was 199.6 kg.

A transmission characterisation of the full setup (including the sliding inlet-system for the APi-TOF) was performed in the laboratory before the campaign. The transmission of ions between 100 Th and 400 Th was between 1–2%, and for ions >500 Th the transmission was below 0.5% (Fig. S3).

## 2.2 Airborne particle measurements

In the second airplane we installed one of these two setups at different stages of the campaign: The first setup was used for the first 4 flight days, and consisted of an ultrafine Condensation Particle Counter (uCPC, model TSI-3776; measuring >3 nm particles), a Scanning Mobility Particle Sizer (SMPS, Wang and Flagan, 1990; measuring a size range 10–400 nm), and a Particle Size Magnifier (PSM, Vanhanen et al., 2011) together with a TSI 3010 CPC (measuring particles with a diameter > 1.5 nm). The second setup consisted of a Neutral cluster and Air Ion Spectrometer (NAIS, Airel Ltd., Mirme and Mirme, 2013) for
the last 2 flight days, measuring neutral particles between 2–40 nm and ions between 0.8–40 nm. Detailed information about the two setups can be found in Lampilahti et al. (2021) and Leino et al. (2019).

## 2.3 Flight trajectories and profiling

We conducted a total of 16 flights (with both aircrafts) of ~2.5 h duration for each flight between 2 May 2017 and 16 May
2017. When possible, three flights per day were performed to capture the development of the boundary layer and its change in particle number concentration and ion composition within one day. Therefore, we conducted an early morning flight between 3:00–6:00 EET (UTC + 2), to capture the stable boundary layer and residual layer; a morning flight between 8:00–11:00 EET, when the mixing of the boundary layer started but was not fully developed; and an afternoon flight between 13:00–16:00 EET,





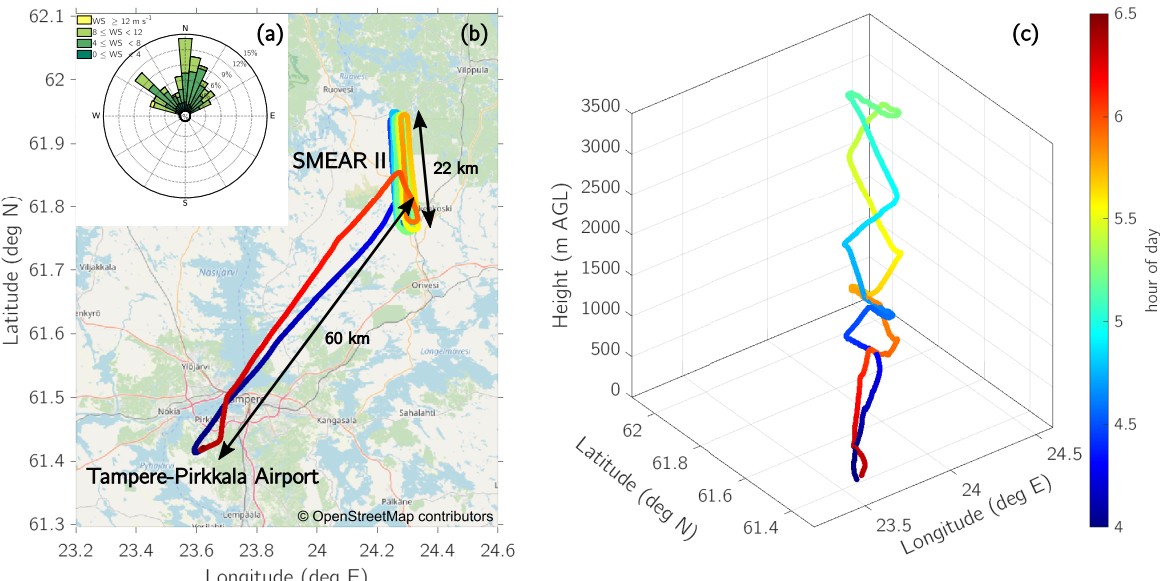

**Figure 2.** (a) Wind rose of the measured wind direction and wind speed from SMEAR II station at 125 m height, including the data from all six days when airborne measurements were conducted. (b) Flight route from the Tampere-Pirkkala Airport (Finland) to SMEAR II station. The depicted route is from a measurement flight on 2 May 2017 during early morning (04:00–06:30 EET). The distance from the airport to SMEAR II station is ~60 km, the length of one long leg at the SMEAR II station is ~22 km. (c) Profile of the same flight as depicted in (b) but including the height. The colouring of the trajectories is represents the hour of the day. ©OpenStreetMap contributors 2021. Distributed under the Open Data Commons Open Database License (ODbL) v1.0.

when the maximum of the boundary layer height was reached. We performed the vertical profiling of the measurements on
fixed altitudes (and pressure, respectively) in ~10 min legs, which enabled a broad vertical profiling of the ion composition. In
this way, we were able to accumulate sufficient amount of data with the APi-TOF despite its pressure sensitivity. The measurement times and height levels of all measurement flights can be determined from Table S1. For simplicity, the height levels in
figures and tables are rounded to the nearest hundred.

The airborne measurements were conducted in five legs per flight between 100 and 3200 m above ground level (agl). For
each height level, the aircraft circled around the SMEAR II station for ~10 min before descending or ascending to the next
level. Figure 2b shows one example of a horizontal trajectory during 2 May 2017 early morning flight. The vertical profile is
depicted in Fig. 2c. For a more direct comparison with ground-based measurements, in all the following instances we refer to
altitude agl from the SMEAR II station (180 m asl).




The flight route started and ended at the Tampere-Pirkkala airport. It took 20–30 min per flight to the measurement site. The vertical profiling was done once arrived at the SMEAR II station (Fig. 2). The two aircraft were flying at the same time but in a staggered formation and at slightly different heights, to avoid contamination. The aircraft with the particle measurements was on the right-hand side of the APi-TOF aircraft. Generally, the flight speed was 130 km h$^{-1}$ but was increased to 150 km h$^{-1}$ depending on flight conditions. Thus, a 1 min flight time was capturing between 2200 and 2500 m.

## 2.4 Ground measurements

A second APi-TOF was deployed on a tower at the SMEAR II station at a 35 m height, i.e., 16 m above the forest canopy (estimated to be at ~19 m height in 2017 from the 2012 reported canopy height at ~17.5 m (Bäck et al., 2012) from which further average growth of 30 cm per year was added). The APi-TOF was equipped with a switching inlet-system, enabling alternating measurements of either natural ions or neutral molecules by using chemical ionisation (CI) with nitric acid (CI-APi-TOF, Jokinen et al., 2012). The cycle between the modes was 3 min for natural ions and 3 min for the CI-mode. The APi-TOF measured in the same polarity (negative) mode as the flying APi-TOF throughout the campaign. To have a comparable dataset between the tower and airborne measurements, we accumulated 10 min of ambient ion data from the APi-TOF in the tower, collected within a 20 min time window when the aircraft was flying at the lowest level (at ~100 m height) above the SMEAR II station. To assure comparability of both mass spectrometers, the flying setup was deployed in the tower next to the switching (CI-)APi-TOF for 3 weeks after the campaign from 19 May to 8 June 2017. The comparison shows that the detection of smaller masses differed slightly, while the higher mass range (>400 Th) was comparable between both instruments (Fig. S4).

For determination of the boundary layer height, we used the attenuated backscatter coefficient and vertical velocity measured with a Doppler lidar at the SMEAR II station. A more detailed description of the measurements and data treatment can be found in Manninen et al. (2018). Additionally, we used the temperature and water vapour profile from the flight measurements.

Particle measurements were taken from a differential mobility particle sizer (DMPS, Aalto et al., 2001) for particles with a diameter 3–1000 nm, and from a Neutral cluster and Air ion Spectrometer (NAIS, Airel Ltd., Mirme and Mirme, 2013) for particles 2–42 nm and for ions 0.8–42 nm.

## 2.5 Analysis and data processing

The APi-TOF data were analysed using the tofTools toolbox, developed by Junninen et al. (2010) in Matlab. As mentioned in the previous section, the data acquisition of only 10 min per leg is minimal for accumulating sufficient signal of naturally charged ions, as the instrument only captures about 1% of the total atmospheric ions (see section 2.1) and the usual ion concentration is 200–2500 pairs cm$^{-3}$ (positive and negative pairs) (Hirsikko et al., 2011).

During the flight, interference due to radio and mobile phone signals caused additional noise in the data acquisition. For data processing, the noise had to be removed before averaging the spectrum to obtain reliable peaks for high resolution analysis



of their chemical composition. Therefore, we used a noise-removal filter, implemented in tofTools; a threshold of 0.01 counts
per second (cps) was set for a 1-second acquisition period, defining all incoming signal with $\leq 0.01$ cps as noise. After noise
removal, the data yielded in a much clearer spectrum (Fig. S5).

Since the highest signal in the APi-TOF during daytime was composed of sulfuric acid ($H_2SO_4$) peaks, such as sulfuric
acid monomer ($HSO_4^-$, bisulfate ion), sulfuric acid dimer (($H_2SO_4$)$HSO_4^-$) and sulfuric acid trimer (($H_2SO_4$)$_2HSO_4^-$), their
vertical behaviour could be captured with a 2 min resolution. This enabled us to estimate the vertical sulfuric acid concentra-
tion, as performed in Beck et al. (2021b), during ascents and descents. However, such vertical concentration profiles of other
compounds were not possible, as their signal was too low and was primarily lost to noise.

An estimation of the sulfuric acid concentration based on ion signals in APi-TOF was tested with the intercomparison period
mentioned in section 2.4 of both mass spectrometers at the SMEAR II station. A comparison with measured sulfuric acid con-
centrations revealed that the daytime sulfuric acid concentration was well estimated with this method (Beck et al., 2021b, Fig.
S6). However, it should be noted that the changes in pressure during the flight measurements may create additional sources of
error due to a change in transmission and fragmentation within the APi-TOF.

## 3 Results and discussion

### 3.1 Campaign overview

Figure 3 provides an overview of the ground measurements during the flight campaign. Panel (f) indicates the days when
airborne measurements were conducted, including the instrumentation installed on the aerosol aircraft. The particle size dis-
tribution and negative ion size distribution (Fig. 3a, b) show that on each flight day, NPF was observed close to the ground.
The temperatures varied between -5 and 18°C (Fig. 3c). Since the CI-APi-TOF at the tower at SMEAR II station occasionally
malfunctioned, we used the sulfuric acid proxy based on Dada et al. (2020) to estimate the sulfuric acid concentration at the
ground (Fig. 3e). The proxy correlates with the measured sulfuric acid concentration (during the times when CI-APi-TOF was
functioning) by 0.79 (Pearson correlation coefficient). On days with NPF, the maximum sulfuric acid concentration reached
$7 \cdot 10^5$ to $3 \cdot 10^6$ molecules cm$^{-3}$. The temperature, relative humidity and potential temperature profiles of all flights are shown
in Fig. S7 to S12.

The aircraft measurements were designed to capture different layers of the boundary layer and the free troposphere. There-
fore, we will briefly summarise the boundary layer development as it occurred during our measurement period. As shown in
the schematic in Fig. 4 based on Stull (1988), the classical boundary layer (BL) structure consists of several sub-layers that
are formed due to a diurnal variation of temperature and heat transfer. In the studied region and during the time of the year,
when the measurements were conducted, the following description of the BL structure can be assumed to apply for most days






**Figure 3.** Ground measurements at Hyytiälä, SMEAR II station from 2 May to 16 May 2017. (a) Particle size distribution of particles with a mobility diameter of 3 nm to 1000 nm. (b) Negative ion size distribution of ions in the size range 0.8–20 nm. (c) Temperature and UV-B radiation. (d) $O_3$ and $NO_x$ concentration. (e) Sulfuric acid concentration measured (red dots) and calculated with a proxy based on Dada et al. (2020). (f) Data availability of the flight campaign distinguishing between early morning, morning and afternoon flight, as well as the instrumentation setup on both airplanes. The instrumentation in the aerosol measurement airplane was changed between 5 May and 12 May 2017 from PSM, uCPC and SMPS to a NAIS. The exact times when the measurements were conducted are shown in Table S1.

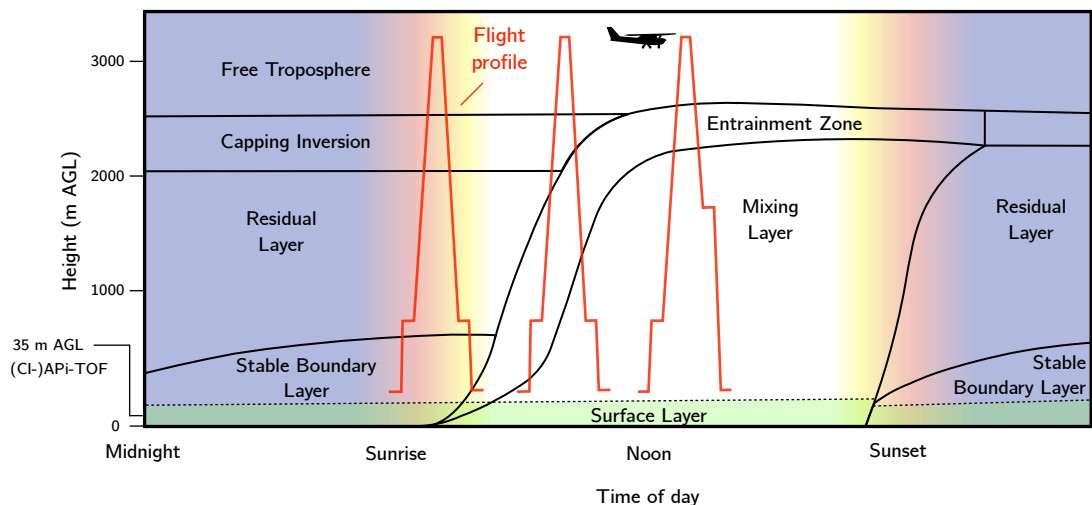

**Figure 4.** Schematic of the boundary layer development based on Stull (1988) with example height profiles of airborne measurements. The actual layer heights may vary from the values on the values depicted on the vertical axis. The tower with the (CI-)APi-TOF is 35 m tall and reaches above the canopy (19 m agl in 2017).

with similar weather conditions. During daytime, a mixing layer (ML) is formed via convective turbulence. This is caused by heating of the surface through radiation, causing warmer air at the lower levels as updrafts to rise and cool adiabatically. Once the air parcels cool down and reach a lower temperature than the surrounding air, they start descending due to their higher
density. This vertical exchange causes turbulent mixing of air parcels throughout the whole BL, increasing the vertical extent of the BL. The BL usually reaches its maximum height in the afternoon. The mixing ceases once the buoyancy is lost due to cooling or reaching an inversion. However, if the moisture of the air parcel condenses and forms a cloud, the latent heat creates additional turbulence (Oke, 1987).

The upper 10% of the ML at the transition to the free troposphere (FT) is called the entrainment zone. The heat flux is reversed due to an inversion within the entrainment zone. However, some air parcels that manage to go past the inversion, are eventually repulsed and transport the warmer air downwards. The entrainment zone is characterised by a significant change in the water vapour content, potential temperature and sensible heat flux between the ML and FT (Stull, 1988; Oke, 1987).





With the setting sun, the turbulence and mixing of the BL start to cease (Lothon et al., 2014; Darbieu et al., 2015). However, the previously mixed air from the ML remains. This leftover layer, which contains the residues of the ML from the previous day, is called a residual layer (RL). The RL persists until the early morning hours when it is mixed again with the ML. Usually, the RL is not attached to the ground, as there is another layer in between. Due to the cooling of the ground during night-time, a very stable layer is formed in the lowest level of the BL, called a stable (nocturnal) boundary layer (SBL). The lowest 10% of

the BL, which is primarily influenced by small-scale turbulences, is called the surface layer. The transition between the SBL and RL is not as distinct as that in the entrainment zone (Oke, 1987).

During night-time with ceasing turbulent mixing, the entrainment zone starts to transfer into a capping inversion. The capping inversion forms a border between the RL and FT (Stull, 1988). The term free troposphere (FT), called sometimes a "free

atmosphere", comprises the air between the top of the BL and tropopause. In contrast to the BL, both friction and turbulence due to heat transfer are negligible in the FT. Instead, the air mass transport in the FT is determined primarily by the synoptic, large-scale weather patterns (Holton and Hakim, 2013).

### 3.2   Case study 1: Diurnal development of ion composition in the boundary layer

For the first case study, we chose 2 May 2017, since this day exhibited a classic development of the BL. Furthermore, we were

able to capture NPF in the upper part of the ML during the measurements. The NPF event has been discussed and published by Lampilahti et al. (2021). With this case study, we aim to gain insight into the ion composition and how it developed in the different layers and throughout the day before and during the NPF event.

The attenuated backscatter coefficient and vertical radial velocity from the Doppler lidar in Fig. 5 show the BL develop-

ment on 2 May 2017. During morning hours, the RL is visible up to ~1700 m (Fig. 5a). The FT begins at 2100 m, as can be determined from the water vapour profile in Fig. S13; the water vapour drops from 4 to <1 g kg$^{-1}$ within the entrainment zone (1700–2100 m). With incoming radiation during morning hours, the ML slowly develops and increases with height, as apparent from the vertical radial velocity profile, which reached its maximum of 1.5 m s$^{-1}$ around 15 EET (Fig. 5b) at the top of the ML (~2000 m), where a cloud layer also formed (Fig. 5a). With the airborne measurements, we were able to measure

the ion composition within the SBL, RL and FT during early morning hours and in the fully developed ML and FT during the afternoon. The ion composition of each layer will be discussed in the following subsections.

### 3.2.1   Vertical profile at sunrise: Ion composition of the stable boundary layer, residual layer and free troposphere

During the first measurement flight on 2 May 2017, we captured the ion composition of the SBL and RL, and the FT before

the sunlight intensity was high enough for stronger photo-oxidation (UV-B < 0.5 Wm$^{-2}$, Fig. 6a.2 and Fig. 3c). The profiling around sunrise (04:14 EET, Table S2) was performed between 04:10–06:10 EET at three different height levels (100 m,

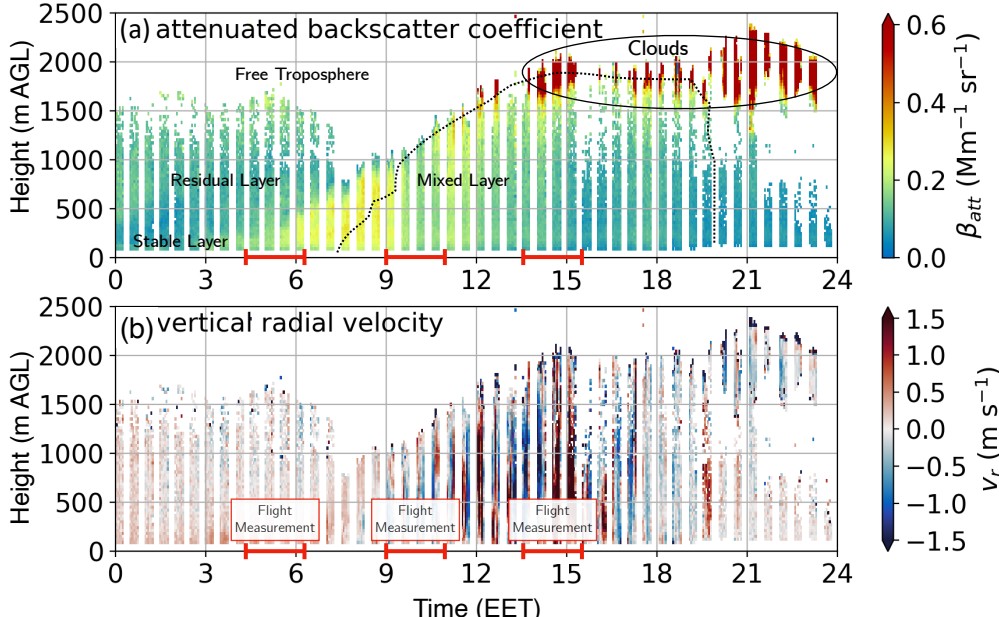

**Figure 5.** Doppler lidar measurement at Hyytiälä, SMEAR II station from 2 May 2017. (a) Attenuated backscatter coefficient and (b) vertical radial velocity measured on 2 May 2017 at SMEAR II station. The measurements were used to estimate the height and vertical extent of the different layers of the boundary layer, such as stable layer, residual layer, mixed layer and free troposphere. The red line on the time axis indicates the airborne measurement periods. The vertical radial velocity (b) is showing the vertical mixing and the resulting increase of the boundary layer height. During early morning flight (04:00–06:00 EET) the airborne captured the stable (boundary) layer, residual layer and the free troposphere. The clouds are visible in the attenuated backscatter coefficient in panel (a). The measurements during morning from 09:00–10:15 EET were conducted only within the mixing layer (< 1000 m), in the afternoon flight, in addition to the mixing layer, the free troposphere was also measured.

700 m, and 3200 m, Fig. 6a.1). In this section, we will describe the ion composition of the different layers: the SBL, RL and FT.

During early morning hours in the SBL at 35 m, the spectrum in Fig. 6b shows ions dominated by monoterpene HOM monomers (mass range of 250–450 Th), and HOM dimers (450–600 Th) (see Bianchi et al., 2019 for the definition of HOM monomer and dimer). The majority of the HOM we detected with the APi-TOF are charged by natural clustering with $NO_3^-$ or $HSO_4^-$ (see also Table A1). Additionally, it should be noted that the total HOM signal over the 300–600 Th range is amplified by a factor of 4 in Fig. 6 (and following figures) to make it more visible.


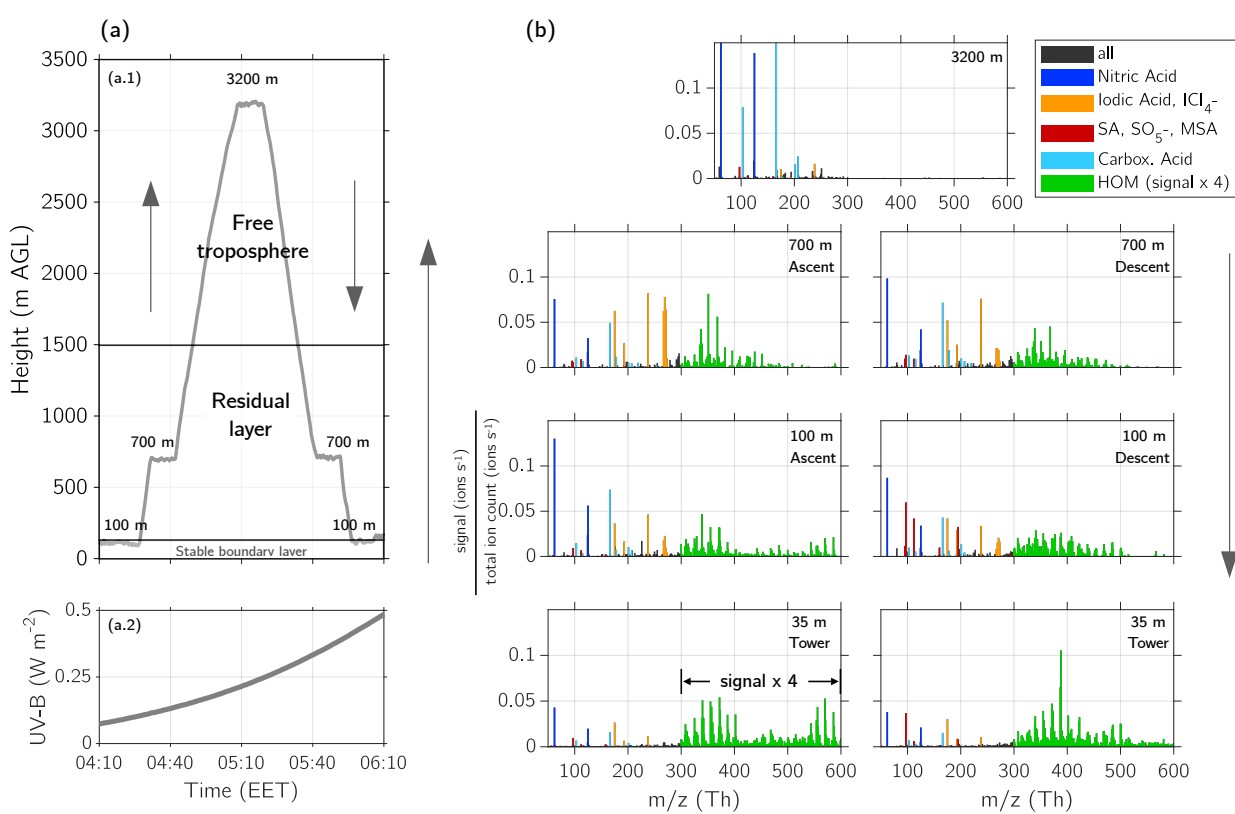

**Figure 6.** Vertical profile of ions above Hyytiälä on 2 May 2017 04:00–06:00 EET. (a.1) Height profile of the flight measurements and indication of the different layers (stable boundary layer (SBL), residual layer (RL), and free troposphere (FT)). (a.2) UV-B radiation measured on the ground at SMEAR II station during the flight. (b) Mass spectra measured with APi-TOF at different heights. The altitude where the mass spectra were measured is indicated in the figures and can be compared with the height profile of panel (a). The lowest mass spectra show the measurements from the tower at SMEAR II station at 35 m height, while the presented data is covering the time period of the lowest level measurements at 100 m (04:10–04:20 EET and 05:55–06:05 EET) to be comparable. The signal of the HOM dominated ions (green) between 300–600 Th is multiplied by factor 4 to make the pattern more visible. A peak list of the different layers can be found in Table A1. The colour code shows the composition of the compounds: nitric acid (blue), iodic acid and iodine-chlorine (yellow), sulfuric acid (SA), $SO_5^-$ and MSA (red), carboxylic acids (cyan), and HOM dominated ions (green).





In addition to HOM, the spectrum shows nitric acid ($NO_3^-$ and $HNO_3NO_3^-$), followed by some signals of deprotonated sulfuric acid ($HSO_4^-$, bisulfate ion), iodic acid ($IO_3^-$) and malonic acid ($C_3H_3O_4^-$). The exact ion composition and their mass are presented in Table A1, altogether we are able to identify 96 ion compositions. The observed ions in the SBL agree with observations of previous studies performed at the SMEAR II station (Bianchi et al., 2017; Zha et al., 2018; Yan et al., 2018; Ehn et al., 2010).


The flight measurements show that the spectral pattern at 100 m differed from that in the lowest level of 35 m (SBL); while HOM monomers were detected at both levels, only low concentrations of HOM dimers were observed at 100 m. Furthermore, compared to the SBL, this layer showed higher abundancies of nitric acid, carboxylic acids (malonic and maleic acid, see Table A1) and iodic acid. What distinctly differentiated the layer at 100 m from the SBL is the $ICl_4^-$ ion. This same ion was visible in

the next layer at 700 m, dominating the spectrum together with iodic acid. At 700 m, no HOM dimers were visible, but instead there were HOM monomers. As discussed previously, the 700 m height level can be attributed to the RL, whereas the layer at 100 m was located likely between the SBL and RL. This could also explain the presence of both organics from the SBL below and the iodine/chlorine compounds from the RL above. The 72 h back trajectories calculated from HYSPLIT (Fig. S14a, (Stein et al., 2015; Rolph et al., 2017)) show that the air mass of the RL originates from the open ocean beyond Norway and had been

in contact with the marine boundary layer at the Norwegian Sea, possibly explaining the origin of halogen compounds within the RL.

Most of the detected ions in the FT during early morning (Fig. 6b, 3200 m) were nitric acid, malonic acid, sulfuric acid, iodic acid and some unidentified compounds. No significant signal >300 Th was visible in the FT.


During the descent, with an increasing radiation intensity (0.25 $Wm^{-2}$ < UV-B < 0.5 $Wm^{-2}$), the ion pattern differed from that in the ascent measurements (UV-B < 0.25 $Wm^{-2}$). The sulfuric acid and $SO_5^-$ signal increased, and the HOM pattern changed, showing less dimers at both 100 m and 35 m.

### 3.2.2    Vertical profile during afternoon: Ion composition of the mixing layer and free troposphere

The measurement flights during afternoon on 2 May 2017 captured the ion composition of the ML and the FT between 13:35–15:30 EET. The profiling was performed at five altitude levels, four within the ML (100 m, 400 m, 700 m, and 1500 m) and one in the FT (2700 m) (Fig. 7a.1). During this measurement flight, we additionally observed NPF in the upper ML, as reported by Lampilahti et al. (2021) and discussed in more detail in section 3.2.3.

Within the ML, we captured the vertical profile of HOM. From the lowest level above the canopy (35 m) up to the highest measured level of the ML (1500 m), the spectra of the APi-TOF showed the same HOM pattern, consisting mainly of HOM monomers throughout the whole ML. High-resolution analysis of the different peaks shows that the same compounds detected close to the ground were also observed in the other measured heights (Table A1). This leads us to conclude that the VOC from



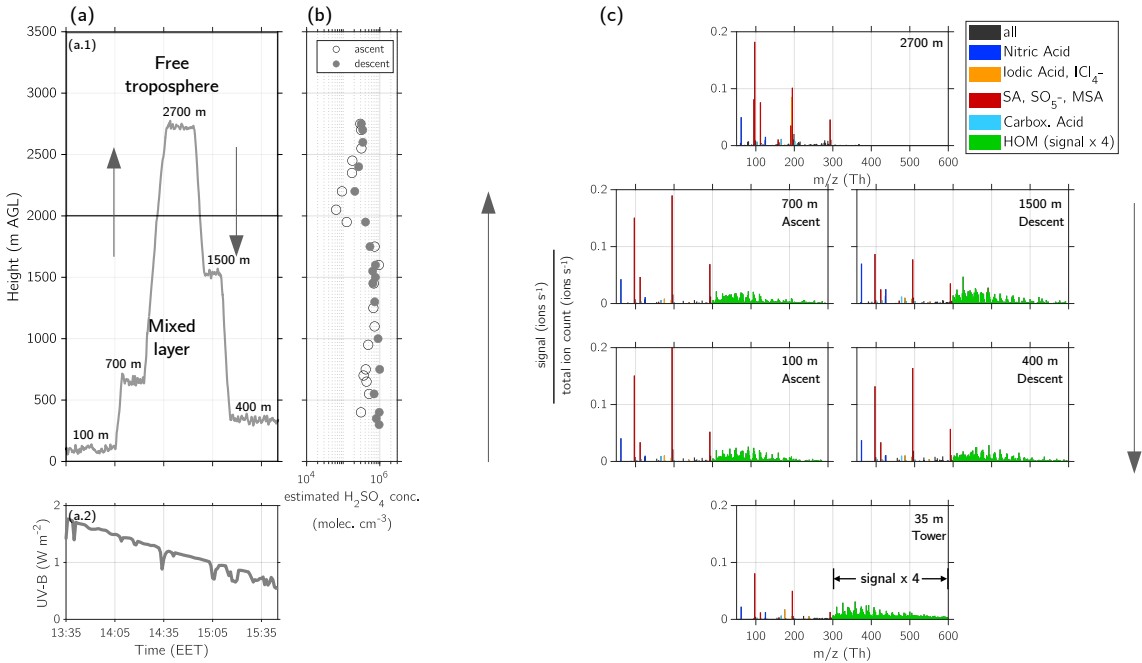

**Figure 7.** Vertical profile of ions above Hyytiälä on 2 May 2017 13:30–15:30 EET. (a.1) Height profile of the flight measurements and indication of the different layers (mixing layer (ML) and free troposphere (FT)). (a.2) UV-B radiation measured on the ground at SMEAR II station during the flight. (b) Estimated sulfuric acid concentration in molecules cm$^{-3}$ during ascent (13:15–14:40 EET) and descent (14:40–15:30 EET), calculated based on (Beck et al., 2021b). (c) Mass spectra measured with APi-TOF at different heights. The altitudes, at which the mass spectra were measured, are indicated in the figures and can be compared with the height profile of panel (a). The lowest mass spectrum shows the measurements from the tower at SMEAR II station at 35 m height. The presented data is an average from 14:00–16:00 EET. The signal of the HOM dominated ions (green) between 300–600 Th is multiplied by factor 4 to make the pattern more visible. A peak list of the ML and the FT can be found in Table A1. The legend shows the colouring for compounds containing nitric acid (blue), iodic acid and iodine-chlorine (yellow), sulfuric acid (SA), $SO_5^-$ and methanesulfonic acid (MSA) (red), carboxylic acids (cyan) and HOM dominated ions (green).





the lowest levels (such as monoterpenes) were vertically transported throughout the whole BL, where they oxidised to form
HOM. Due to the continuous mixing of air between the different altitudes, the HOM compounds detected on the ground could
therefore also be produced at higher altitudes (see Fig. 5b).

In the study of Kulmala et al. (2013) a representative HOM compound at the SMEAR II station measured with CI-APi-TOF
was at unit mass 339, identified to be $C_{10}H_{15}O_8N \cdot NO_3^-$. This ion was visible in the all layers within the boundary layer (SBL,
RL and ML) (Table A1), keeping in mind that in our observations the HOM $C_{10}H_{15}O_8N$ is naturally charged by clustering
with the $NO_3^-$ ion, as we did not use a chemical ionisation.

Unlike during the early morning hours, the spectrum in the afternoon is dominated by the sulfuric acid monomer, dimer
and trimer. Furthermore, we measured nitric acid, $SO_5^-$ and some iodic acid (Table A1). The observed ions of the ML agree
with previous observations of naturally charged ions at the SMEAR II station, e.g. by Ehn et al. (2010) and Bianchi et al. (2017).

In the FT, most of the detected ions were sulfuric acid monomers, dimers and trimers as well as $SO_5^-$ and nitric acid. Unlike
the ML, the FT also contained MSA monomer and dimer, whereas HOM were not abundant at all. According to the HYSPLIT
back trajectories (Fig. S15), the air mass measured in the afternoon in the FT had a marine origin, explaining the presence of
MSA.

### 3.2.3   New particle formation in the upper mixing layer

A study by Lampilahti et al. (2021) suggests that on 2 May 2017 NPF took place in a layer between the RL and the FT at
around 2000 m altitude. During the afternoon flight the ML reached this altitude and the layer of newly formed particles was
mixed down (see also Fig. 3a, b).

Our ion measurements of this NPF event provide additional insight into the ion composition in the ML. Using the method
introduced previously (Beck et al., 2021b), we estimated sulfuric acid concentration in the ML and FT (Fig. 7b and 8) based
on sulfuric acid monomer, dimer, and trimer concentrations and condensation sink. The estimated sulfuric acid concentration
during ascent between 500 m and 1700 m (ML) varied from 3 to $1 \cdot 10^6$ cm$^{-3}$ and dropped down to a minimum of ~$6 \cdot 10^4$
cm$^{-3}$ at 2100 m in the FT (Fig. 7b and 8c). During the descent, the estimated sulfuric acid concentration in the FT tendentially
increased, and the concentration change between the FT and ML was not as drastic as during the ascent. During the descent
in the ML, the sulfuric acid concentration varied between $5 \cdot 10^5$ and $1 \cdot 10^6$ cm$^{-3}$. In general, the sulfuric acid concentration in
the FT was lower by about an order of magnitude compared with the ML (Fig. 8).


At the observed temperatures (-8 to 0°C) and relative humidities (15–50%) in the upper ML, a sulfuric acid concentration of
at least $6 \cdot 10^6$ cm$^{-3}$ would be needed to achieve a nucleation rate of 1 cm$^{-3}$s$^{-1}$ when considering binary sulfuric acid – water

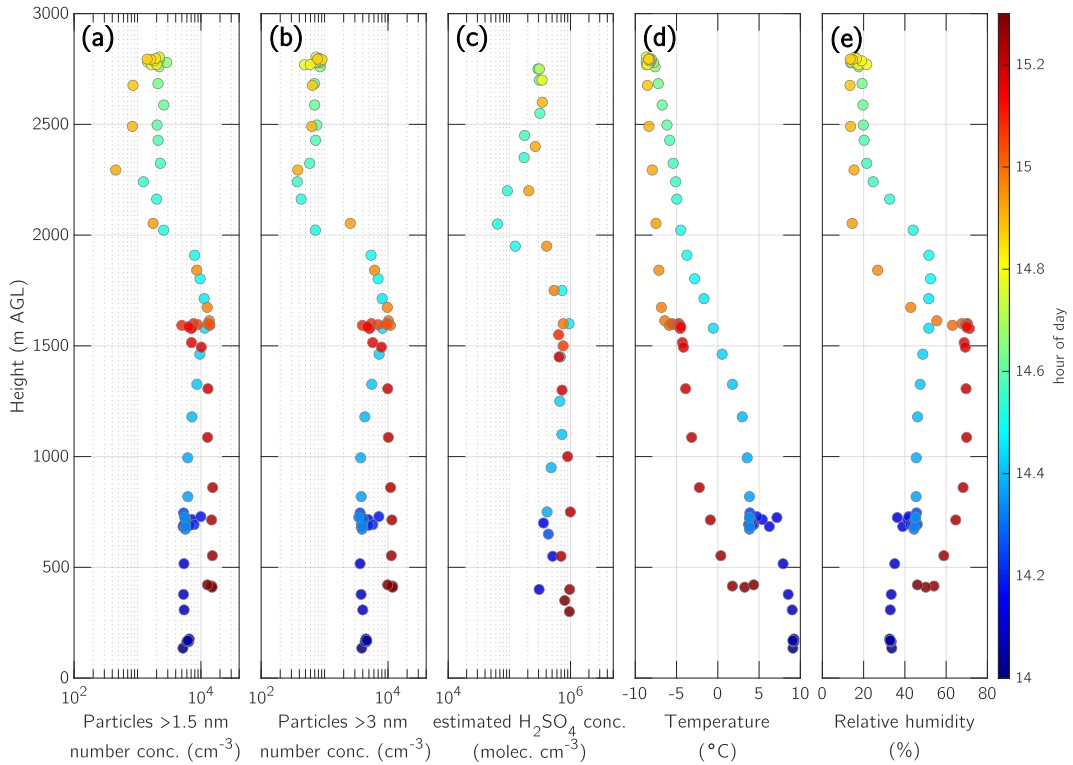

**Figure 8.** : Vertical profile of the afternoon flight from 2 May 2017 14:00–15:20 EET, showing particles >1.5 nm (a), particles >3 nm (b), estimated sulfuric acid ($H_2SO_4$) concentration (c) (Beck et al., 2021b), temperature (d) and relative humidity (e). The colour indicates the hour of the day such that ascent and descent are distinguishable

nucleation (Duplissy et al., 2016). In our observations, the estimated sulfuric acid concentration was $\leq 10^6$ cm$^{-3}$ throughout the ML (Fig. 8c). When the nucleation event was detected at the ground at the SMEAR II station around noon, the nucleation

rate of particles with a diameter of 3 nm was 0.19 cm$^{-3}$s$^{-1}$ at 10°C and at the sulfuric acid concentration of 6·10$^5$ cm$^{-3}$ (Fig. 3), which is insufficient for pure sulfuric acid nucleation (Duplissy et al., 2016).

However, as the spectrum of the APi-TOF shows, there was a high abundancy of other compounds that can initiate and contribute to both clustering and growth, such as HOM and iodine compounds (Fig. 7). While HOM have been recognised to

participate in the NPF processes in Hyytiälä (Lehtipalo et al., 2018), recent findings reported extremely rapid iodine oxoacid particle formation, and iodic acid was observe in both Hyytiälä (Sipilä et al., 2016; He et al., 2021) and Arctic (Beck et al., 2021a; Baccarini et al., 2020) from where the air masses measured here originated from (Fig. S6). Therefore, iodic acid could




have a minor contribution to either NPF or the growth processes, or both of them. Unfortunately, the averaging time of 10 min for each layer was not sufficient to detect clusters having a low concentration. We therefore could not identify a clear nucle-
ation path of the very first clusters, such as for example of pure $H_2SO_4$, $H_2SO_4-NH_3$, $H_2SO_4-DMA$, $H_2SO_4-HOM-NH_3$, $H_2SO_4-HOM-DMA$, or other mechanisms including those involving iodic acid. However, in this case study, the iodate ion ($IO_3^-$) remained low, which indicates a minor contribution from iodic acid at least in the ion-induced nucleation channel. We measured similar levels of sulfuric acid ions and HOM ions throughout the ML (Fig. 7) which suggests that their concentrations were similar within the ML. Additionally, ammonia has a typical lifetime of a few hours to a day and should be present at different levels of ML if present at the ground level, as suggested by earlier studies (Schobesberger et al., 2015; Lehtipalo et al., 2018). In conclusion, we hypothesise that the $H_2SO_4-HOM-NH_3$ multicomponent nucleation mechanism, as postulated for a boreal forest environment based on laboratory experiments (Lehtipalo et al., 2018), may still be the controlling mechanism aloft of Hyytiälä.

## 3.3  Case study 2: Changing ions in the stable boundary layer during sunrise

For the second case study, we chose 16 May 2017 with a focus on the early morning flight. On this day, the change of ions in the SBL during early morning was very pronounced. Additionally, the RL was distinctly different from case study 1, reaching only up to 1000 m (Fig. 9) (in comparison, on 2 May 2017 it reached ~1700 m).

The early morning measurements on 16 May 2017 started at around 03:30 EET and ended at 05:30 EET (Fig. 10). Compared to the first case study (2 May 2017), the sun rose almost 40 min earlier at 03:38 EET (Table S2). Therefore, the starting time of the early morning measurement was accordingly adapted and performed 40 min earlier than during previous measurements, so that the radiation levels (UV-B <0.5 W m$^{-2}$, Fig. 10a.2) were comparable with the other observations. The profiling of the flight was conducted at the following four different altitudes: 100 m (SBL), 700 m (RL), 1500 m, and 2600 m (FT). The SBL was captured twice, during ascent and descent, with a time difference of 1.5 hours (Fig. 10 a.1).

During morning hours, the sulfuric acid concentration typically increases with time (Fig. 3e), which in turn harvests most of the negative charges (the proton affinity of sulfuric acid is low compared to other acids, such as $HNO_3$, $HIO_3$, MSA), reducing the chance to detect other species because they would need to be charged naturally to be detected by APi-TOF. This feature was visible within the 1.5-hour period of the early morning measurements. When starting at the 100 m level within the SBL at 03:35 EET (UV-B <0.1 W m$^{-2}$), the spectrum of ions shows a typical pattern of night-time HOM clustered with $NO_3^-$, including $C_{20}H_{31}O_{13}N \cdot NO_3^-$, $C_{10}H_{15}O_{12}N \cdot NO_3^-$, and $C_{20}H_{32}O_{12}N_2 \cdot NO_3^-$ (Yan et al., 2016). Additionally, nitric acid, iodine compounds, a small amount of sulfuric acid clusters and $ICl_4^-$- are visible. The spectrum at the 100 m level is similar to the one measured in the tower, except that iodic acid and $ICl_4^-$ were not visible near the ground.



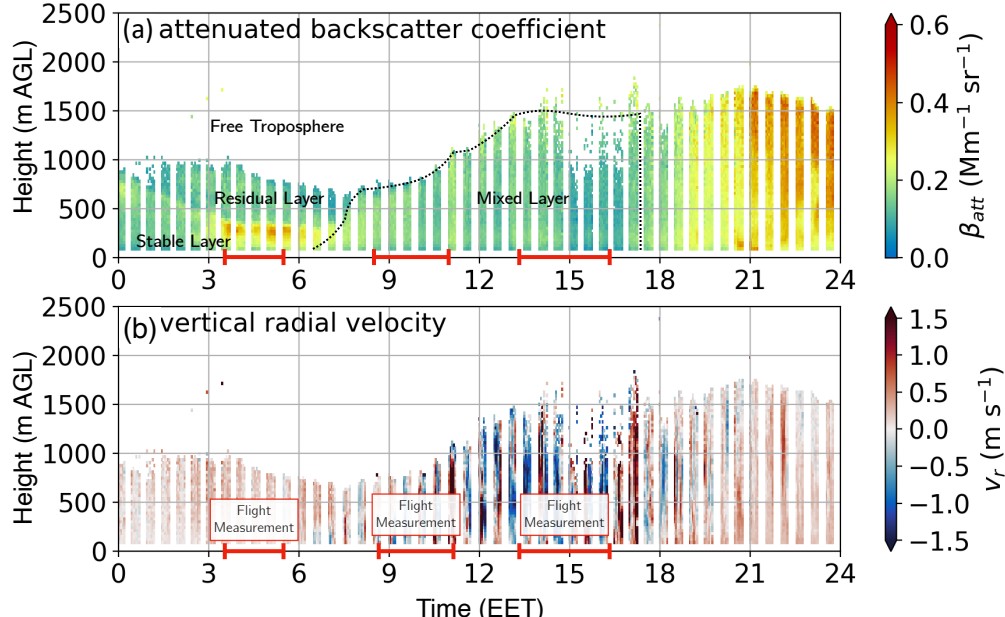

**Figure 9.** Doppler lidar measurement at Hyytiälä, SMEAR II station from 16 May 2017. (a) Attenuated backscatter coefficient and (b) vertical radial velocity measured on 16 May 2017 at SMEAR II station. The measurements were used to estimate the height and vertical extent of the different layers of the boundary layer, such as stable (boundary) layer, residual layer, mixed layer and free troposphere. The red line on the time axis indicates the airborne measurement periods. The vertical radial velocity (b) shows the vertical mixing and the resulting increase of the boundary layer height.

About 1.5 hours later, during the descent (05:10 EET), the UV-B radiation had increased to 0.25 W m$^{-2}$ (Fig. 10a.2). As a result, photochemistry started and sulfuric acid peaks became the most abundant ones in the spectrum, whereas the HOM dimers and iodine compounds disappeared (their charges being transferred to sulfuric acid) and daytime HOM ions became visible, as reported by Yan et al. (2016). Although the time of measurements and radiation intensity on the 16 May 2017 case

is comparable with the 2 May 2017 case (regarding sunrise and time of measurement), the 16 May 2017 case showed a much stronger sulfuric acid signal in the SBL during early morning than what was observed on 2 May 2017. This might simply be because the sulfuric acid concentration on the ground around the time of the descent in the SBL was higher by a factor of five (~3·10$^6$ cm$^{-3}$) on 16 May 2017 compared to that (~6·10$^5$ cm$^{-3}$) on 2 May 2017 (Fig. 3e).

Unlike on 2 May 2017, we did not detect any HOM (Fig. 10) in the RL (700 m) during the early morning flight on 16 May 2017. Instead, the highest signals were malonic acid, maleic acid, and nitric acid, and some MSA and sulfuric acid. The RL





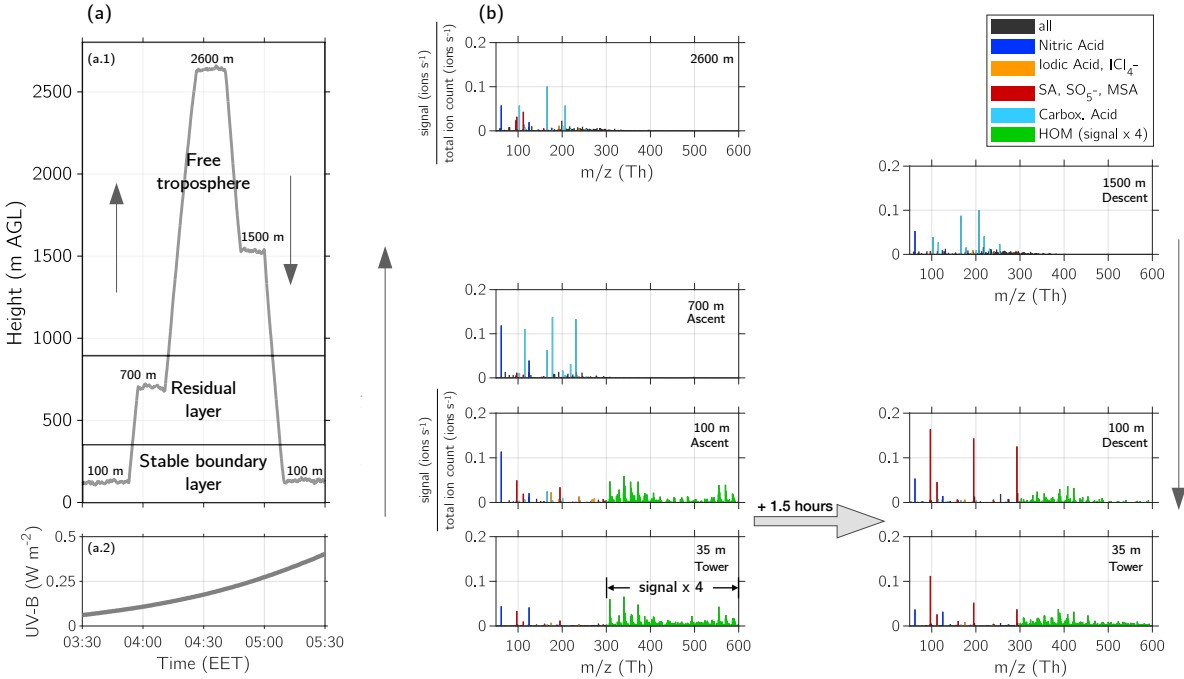

**Figure 10.** Measurements from the night flight on 16 May 2017. Height profile between 03:30–05:30 EET in the morning (a.1) as well as UV-B radiation measured on the ground (a.2). (b) Mass spectra during ascent as well as measurements from the tower, the height levels are indicated in each panel. The layers of the BL with estimation of SBL and RL are indicated in the height profile of panel (a.1). The time difference between the measured mass spectra inside the stable layer at 100 m is 1.5 hours. With upcoming sunlight, the sulfuric acid clusters become the most abundant peaks in the spectrum.

and FT will be discussed in more detail in section 3.4.

### 3.4 Variability of ions in the stable boundary layer and residual layer

When comparing the six measurement days, the highest variation of ions is observed within the RL, while the other layers show considerable similarities between each day. On all the flights, the first profiling was performed at roughly 100 m above SMEAR II. Here, depending on the weather conditions, we captured the SBL with either a strong or less pronounced inversion. The next higher measurement level during the early morning flights was at 700 m (within the RL) before moving on to the highest measurement level ($\leq$ 2500 m) in the FT. In this section, we will discuss the ion composition of those three layers in

more detail.



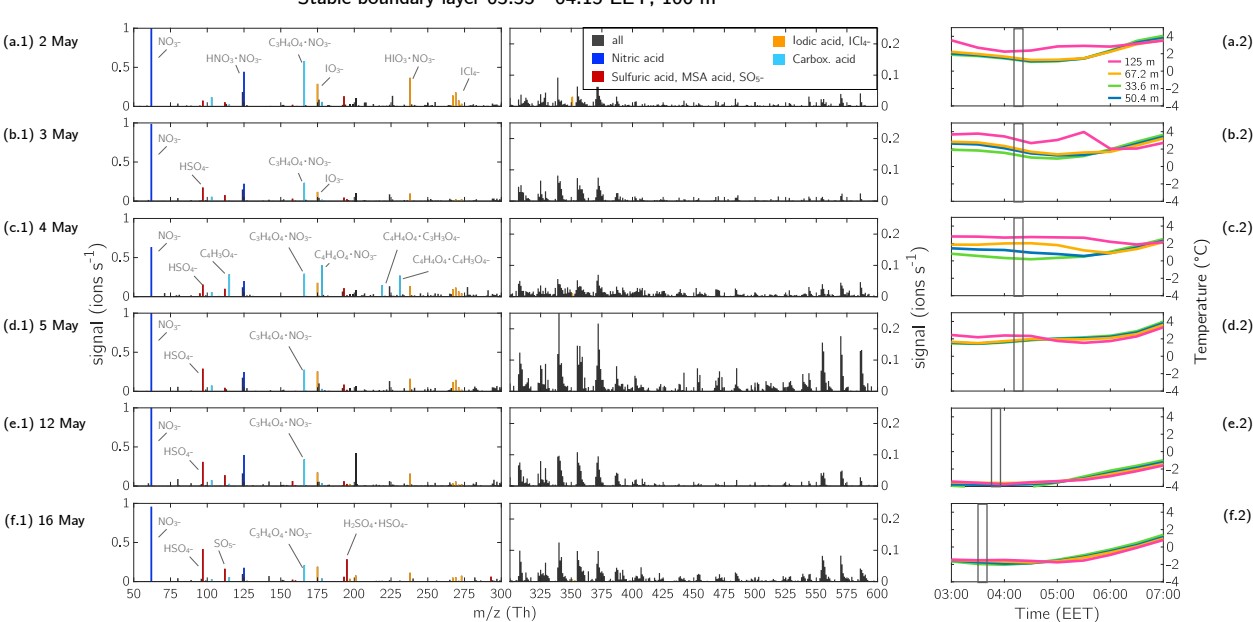

**Figure 11.** Measurements from 100 m above the ground during early morning between 03:45 and 04:15 EET. Ion spectra measured with APi-TOF (a.1–f.1). The mass axis is split at 300 Th such that the limits of the vertical axis could be decreased to make the ions >300 Th more visible. The composition of identified ions with a high signal are noted in the figure. Panels (a.2–f.2) show the temperature profile measured at SMEAR II station at the following four levels: 33.6 m, 50.4 m, 67.2 m and 125 m. The grey box indicates the measurement time of the ions in panel a.1–f.1.

### 3.4.1   The stable boundary layer

The thickness of the SBL varied from day to day, depending on meteorological conditions. In order to determine whether the airborne measurements at ~100 m height were within the SBL, we used the temperature profile of the tower, located at the SMEAR II station. The tower conducts measurements at different height levels from near the ground up to 125 m. For this

purpose, we considered the four uppermost measurements at 33.6 m, 50.4 m, 67.2 m, and 125 m, as these capture the measurements in the tower (35 m) from above the canopy up to the airborne measurements at ~100 m. The temperature development during the flight measurements at the different levels are shown in Fig. 11a.2–f.2. During the flights on 2, 3, and 4 May, the temperature at 125 m was higher than at 67.2 m (Fig. 11 a.2–c.2) by 0.7–1.2°C, showing a rather strong inversion. On 5, 12, and 16 May, the temperature difference between 67.2 and 125 m was only 0.2–0.3°C, so on these days the inversion was not

as clearly pronounced (Fig. 11d.2–f.2).





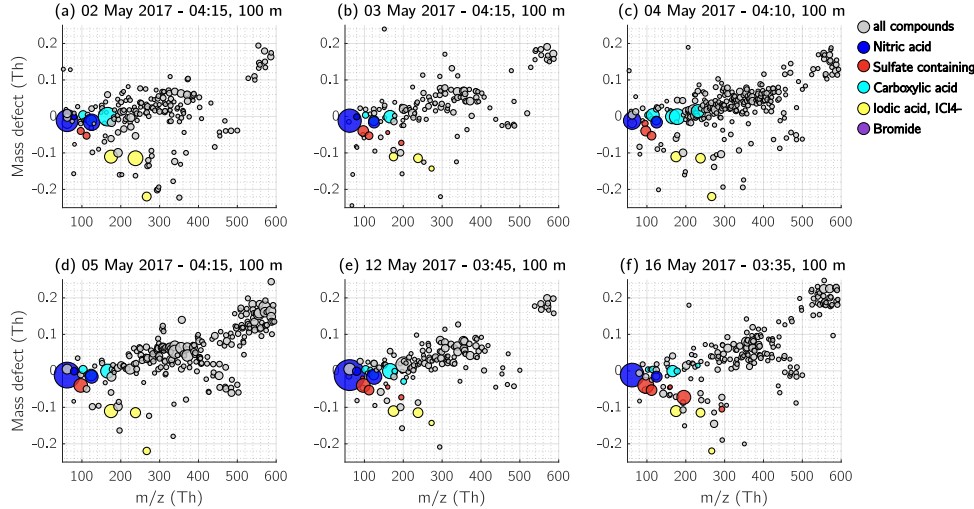

**Figure 12.** Mass defect of the detected ions in the SBL at 100 m height. Each measurement day is shown separately. Nitric acid containing ions are coloured dark blue, and ions and clusters with sulfuric acid, MSA, or both and the the $SO_5^-$ ion are indicated in red. The compounds containing carboxyl functional groups are shown in cyan, and iodic acid and iodine-containing compounds are shown in yellow. The size of the dots corresponds to the measured signal (ions s$^{-1}$).

The ions with the highest signal in the BL were nitric acid (monomer and dimer), carboxylic acids, sulfuric acid, and iodic acid. The $ICl_4^-$ ion, mentioned in case study 1, was also observed on other days (2, 4, 5, 12, and 16 May 2017) (Fig. 11). Some carboxylic acids were visible on every day, mainly malonic acid clustered with nitric acid, but on 4 May 2017 we additionally

observed maleic acid in the SBL (Fig. 11c.1).

In the mass range of >300 Th, the ion pattern of the SBL varied from day to day. On days with a strong inversion (2, 3, and 4 May), the signal of the HOM monomers between 300–450 Th was mainly <0.05 ions s$^{-1}$ and was even lower for HOM dimers (450–600 Th). However, on the other days when the inversion was not as strong (5, 12, and 16 May 2017), the HOM

signal reached up to 0.1 ions s$^{-1}$ or higher, and more HOM dimers were visible. The variation in HOM compounds on each day can also be determined from the mass defect plot in Fig. 12. The ions with a positive mass defect and a mass between 300–450 Th can be attributed to HOM monomers (naturally clustered with $NO_3^-$ or $HSO_4^-$), and the compounds with a mass >450 Th to HOM dimers (naturally clustered with $NO_3^-$ or $HSO_4^-$). Generally, it can be said that we detected HOM on every measurement flight in the SBL at 100 m, but the fraction of HOM monomers and dimers was varying between the days.




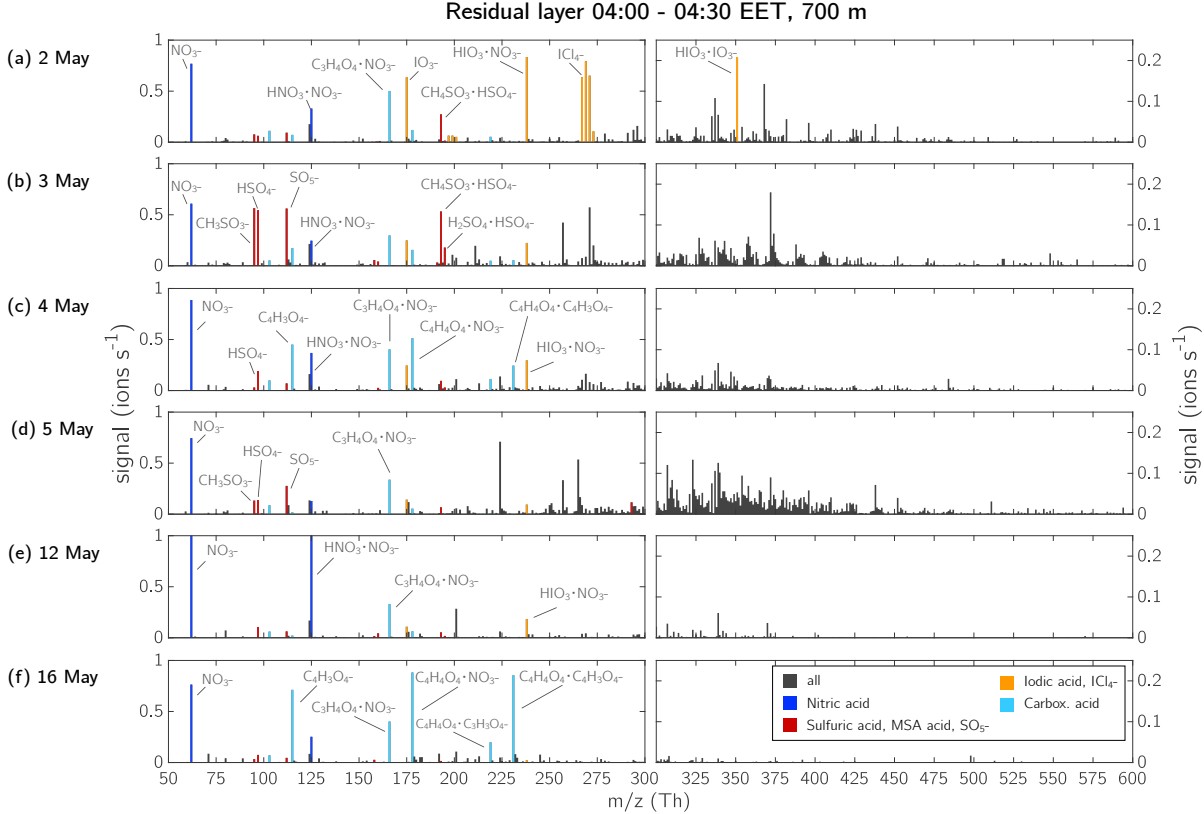

**Figure 13.** Ion spectra inside the RL at 700 m during early morning between 04:00–04:30 EET. Each row is showing one measurement flight; (a) 2 May, (b) 3 May, (c) 4 May, (d) 5 May, (e) 12 May and (f) 16 May 2017. The mass axis is split at 300 Th such that the limits of the vertical axis could be adapted to make the ions >300 Th more visible. The elemental composition of ions with a high signal that could be identified are noted in the figure. The 700 m height level was within the RL on each measurement day. The boundary layer height and development can be determined from Fig. S16 in supplementary material. The nitric acid signal exceeds the limits of the vertical axis on 12 May 2017 in panel (e): $[NO_3^-] = 1.93$ ions s$^{-1}$, $[HNO_3 \cdot NO_3^-] = 1.18$ ions s$^{-1}$.


### 3.4.2 The residual layer

The RL represents a very interesting air mass as this layer is decoupled from the FT and SBL. When mixed with the other layers during daytime, the RL can act as a relevant source of NPF precursors (Lampilahti et al., 2021; Wehner et al., 2010). Here, we show that the RL is either more similar to the SBL (e.g. by containing HOM) or rather shows similarities with the ion composition of the FT.

From the ion spectra of the RL in Fig. 13, it is apparent that the ion composition of the RL varied between each measurement, except for $NO_3^-$ which was abundant every day.

On 2 May 2017, most of the signal was composed of iodic acid, and the $ICl_2^-$ and $ICl_4^-$ ions already mentioned in case study 1 (Figs. 6, 13a, 14a). Some peaks were visible above 300 Th; however most compounds had a negative mass defect (Fig. 14a) and did not seem to be related to any HOM that were detected in the SBL at 100 m (Fig 12a). On the following day, 3 May 2017, the signal in the RL consisted mostly of sulfuric acid and MSA (Fig. 13b and 14b) and HOM monomers were abundant. On 4 May 2017, a mixture of peaks, including nitric acid, carboxylic acids, iodic acid, $ICl_4^-$, sulfuric acid, and HOM monomers were observed (Fig. 13c and 14c).

Comparing all the six measurement days in the RL, the 5 May 2017 measurement showed the highest amount of HOM monomers, while in contrast the two measurement flights one week later, 12 and 16 May, did not show any HOM signal. Instead, >300 Th barely any ions were visible at all and instead the majority of the signal consisted of high amounts of nitric acid (12 May, Fig. 13e, Fig. 14e) and carboxylic acid (16 May, 13f, Fig. 14f).

On days when the air mass of the RL was passing over the Norwegian Sea or the Barents Sea, the spectra contained iodine and chlorine compounds (2, 3, 4, 5, and 12 May 2017). That was not the case on 16 May 2017 when the air mass of the RL originated from the Barents Sea but travelled over land south of the Russian coast for the past 44 hours before arriving at the SMEAR II station (Fig. S14).

### 3.4.3 The free troposphere during early morning

As the FT is decoupled from the BL, its ion composition is determined more by long-range transport rather than local sources. However, it is also subject to daily variability due to photochemistry. In the following section, we discuss the observations of the FT during early morning, morning and afternoon.

During early morning, we observed a large variety of ions in the FT. Figure 15 shows the median of the six measurements of the FT during the campaign, measured between 04:30–05:00 EET. The depicted spectrum is shown in mass unit resolution.



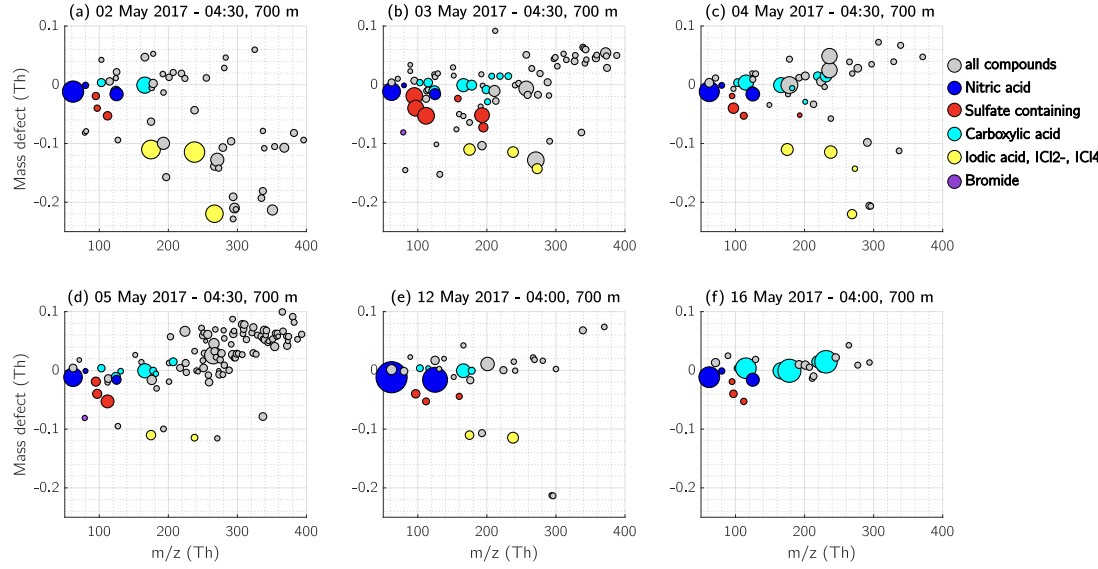

**Figure 14.** Mass defect of the detected ions in the residual layer at 700 m height. Each day of measurement is shown separately. Nitric acid containing ions are coloured dark blue, ions and clusters with sulfuric acid, MSA, or both, and the $SO_5^-$ ion are coloured red. The compounds containing carboxyl functional groups are shown in cyan, and iodic acid and iodine containing compounds in yellow. The size of the dots corresponds to the measured signal (ions s$^{-1}$).

Most of the observed compounds had a mass <250 Th and the highest signal consisted of nitric acid, MSA ($CH_3SO_3^-$), sul-
furic acid, $SO_5^-$, iodic acid, and compounds containing carboxylic functional groups (Fig. 15, Fig. 16 and peak list in Table A2).

  The identified compounds containing carboxyl groups (COOH) were malonic acid, maleic acid, carbonic acid, oxaloacetic
acid, tartonic acid and citric acid (see peak list in Table A2). However, the nomenclature of these acids is only a suggestion, as
we cannot identify the structure of the ions but only their elemental composition. Some of the compounds are clustered with
$HSO_4^-$, $NO_3^-$ and MSA, but they were also observed as oligomers and clustered with another carboxylic acid. On 16 May
2017, the FT showed a large amount of the above-mentioned carboxylic acids. Many of the compounds on that day remained
unidentified, although the majority had a positive mass defect (Fig. 16f) or were spread between -0.05 and +0.05 Th. Since
many of the carboxylic acids were observed at the same time and they tended to have a positive mass defect (e.g., malonic acid
$C_3H_3O_4^-$ with a mass of 103.0037 Th and a mass defect of 0.0037 Th), it is possible that the unidentified compounds were a
variety of molecules clustered with one or several carboxylic acids.

  On 3 and 12 May 2017, we also detected an ion pattern between 350–400 Th shown in Fig. 15c (see also Figs. 16b and 16e).
The peak-pattern consists of a group of ions between 349–355 Th and another group between 393–401 Th. Both ion groups are
apart by 44 Th, which could represent an addition of $CO_2$ (43.99 Th). The low mass defects of those peaks (from -0.1 down to

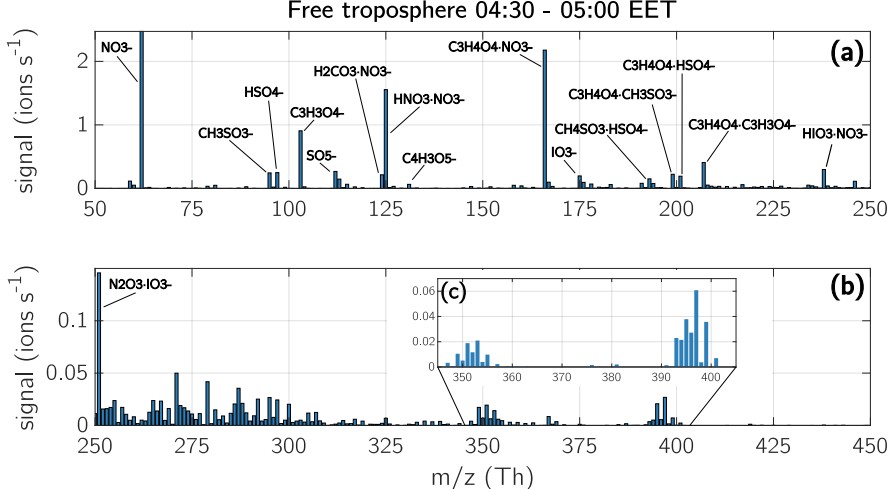

**Figure 15.** Mass spectra of the free troposphere during early morning. The depicted spectra show a median from six measurements conducted during early morning hours between 04:30–05:00 EET in the FT: 2 May (at 3200 m), 3 May (2700 m), 4 May (2700 m), 5 May (2800 m), 12 May (2500 m) and 16 May 2017 (2600 m). The chemical composition of the ions with the highest signal are shown in the spectra. Panel (a) shows the masses from 50–250 Th, panel (b) from 250–450 Th. Please note that the limits of the vertical axis in panel (b) are lower than in panel (a), in order to visualise the compounds with a lower concentration. Panel (c) is a zoom into the mass range from 345 to 405 Th. The averaged signal shown in panel (c) is mainly coming from two measurement days, 3 May and 12 May 2017. The temperature in the FT at measurement height during all six measurement days was between -15 and -2°C.

-0.24 Th; Fig. 16 b and e) and their isotopic pattern (Fig. 15c) let us to hypothesize that they contain halogen compounds, such as chlorine. The air mass on the 3 May 2017 in the FT originated from Svalbard in the Arctic (Fig. S14b), which supports the hypothesis that the ions contain e.g. chlorine. On 12 May 2017, the air masses in the FT travelled over the Russian coast at the Barents Sea and the Kola peninsula (Fig. S14e).

**3.4.4   Free troposphere during daytime**

Similar to the ML, the sulfuric acid concentration in the FT increased during daytime, causing sulfuric acid to take up most of the charges. Therefore, the main compounds we observed during the daytime in the FT were sulfuric acid monomers, dimers and trimers, but also MSA, $SO_5^-$, malonic acid and nitric acid (Fig. 17). The depicted spectra in Fig. 17 are a median of three measurements during midmorning (09:30–10:00 EET) and three measurements during the afternoon (14:30–15:00 EET). The

mass defect of each measurement (Fig. 18) provides further details about the ion composition during morning and afternoon. On 3 and 12 May 2017 there was still an abundance of the unknown compounds between 393–401 Th (discussed in section 3.4.3) during the morning hours (Fig. 18a and d). Unfortunately, no measurements were taken in the FT on these two days





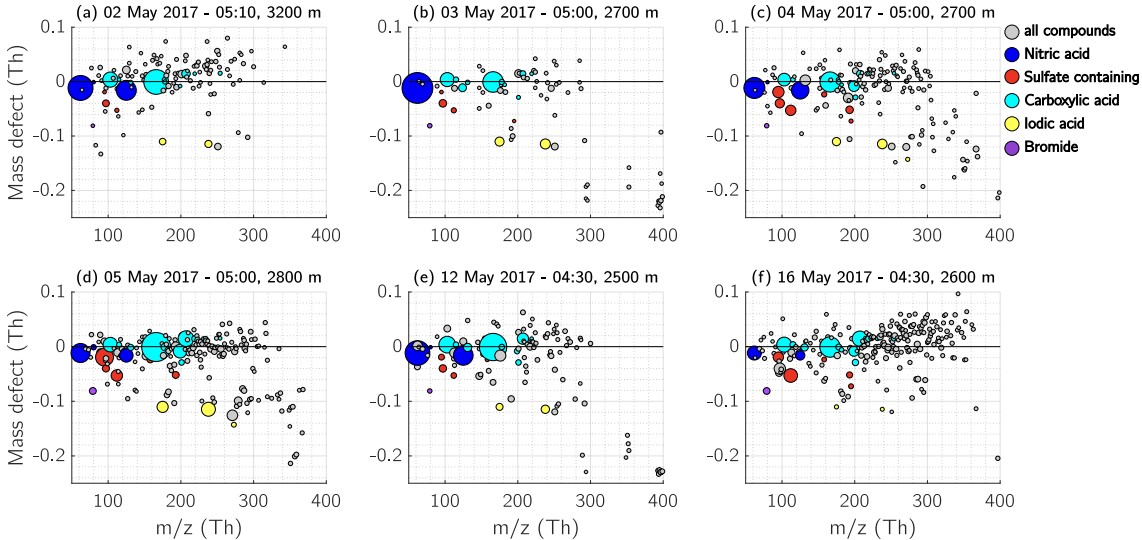

**Figure 16.** Mass defect of the free troposphere during early morning. Each panel from (a) to (f) shows the detected ions in the FT during the six different measurement days. The shown observations were performed during early morning hours between 04:30–05:00 EET. Nitric acid containing ions are coloured dark blue, ions and clusters with sulfuric acid, MSA, or both, and the $SO_5^-$ ion are coloured red. The compounds containing carboxyl functional groups are depicted in cyan, and iodic acid and iodine containing compounds in yellow. The size of the dots corresponds to the measured signal (ions s$^{-1}$).

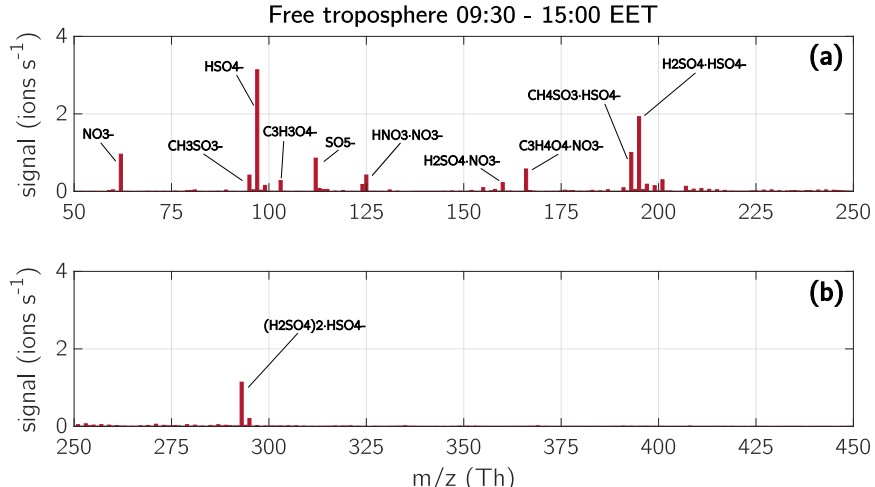

**Figure 17.** Mass spectra of the free troposphere during daytime. The spectra are a median of six measurements in the FT during daytime between 09:30–15:00 EET. The chemical composition of the ions with the highest signal are shown in the spectra. Panel (a) shows the masses from 50–250 Th, panel (b) from 250–450 Th. The temperature in the FT at measurement height during afternoon on all measurement days was between -8 and 3°C.





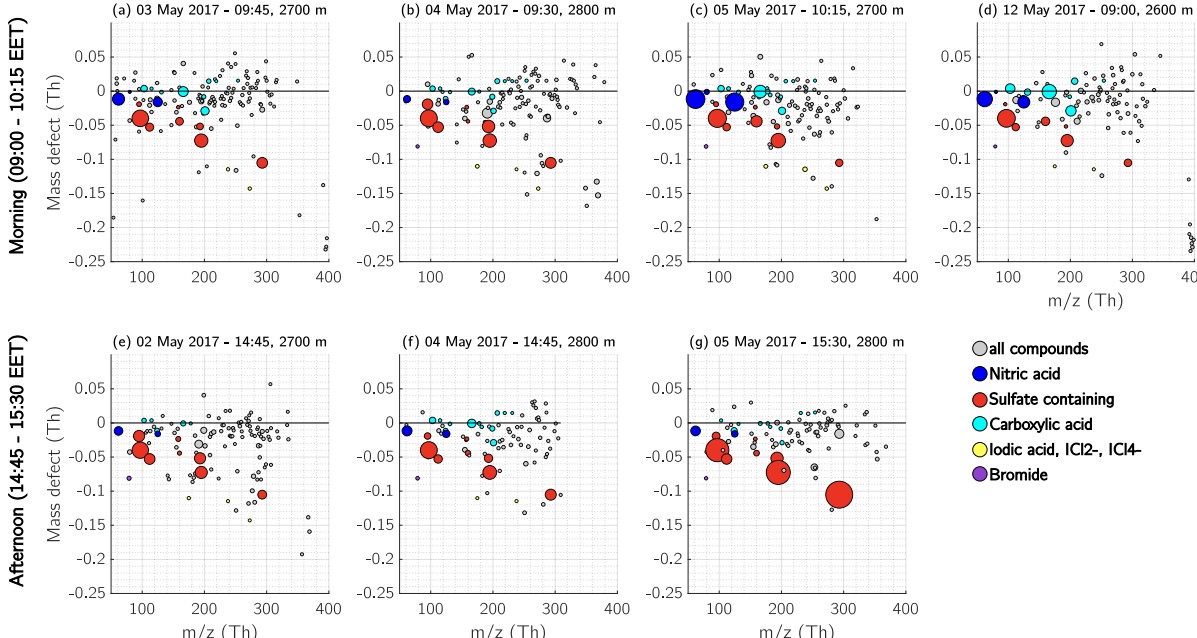

**Figure 18.** Mass defect of the free troposphere during morning and afternoon. Each panel from (a) to (d) shows the detected ions in the FT during morning measurements (between 09:00–10:15 EET), the panels (e) to (g) show the measurements during the afternoons (between 14:45–15:30 EET). Nitric acid containing ions are coloured dark blue, ions and clusters with sulfuric acid, MSA, or both, and the $SO_5^-$ ion are coloured red. The compounds containing carboxyl functional groups are depicted in cyan, and iodic acid and iodine-containing compounds in yellow. The size of the dots corresponds to the measured signal (ions s$^{-1}$).

during afternoon hours.

The carboxylic acids that were detected during the early morning hours were still visible in the morning but showed less signal intensity during afternoon (Fig. 18 e–g). This might be due to the fact that the compounds are not charged as efficiently as in the morning hours, when the sulfuric acid concentration was lower. Although naturally charged carboxylic acid ions were not present and therefore not visible with our APi-TOF mass spectrometer, carboxylic acids might still have been abundant during daytime. Therefore, following the ion composition from early morning to the afternoon, we observed that the carboxylic

acids were less visible during afternoon, and nitric acid decreased in its signal intensity from early morning to the afternoon, and the majority of the ions during daytime is comprised of sulfuric acid.





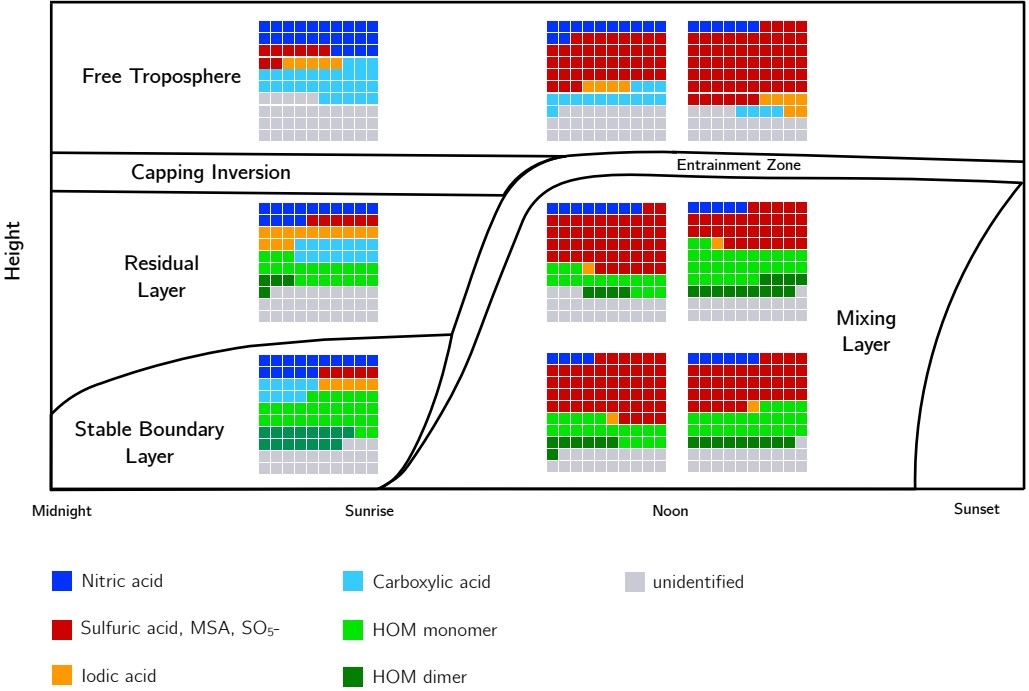

**Figure 19.** Schematic of ions observed within the boundary layer and free troposphere. The waffle plots show the fraction of different compounds, such as nitric acid (dark blue), sulfuric acid, MSA, and $SO_5^-$ (red), iodic acid (yellow), carboxylic acid (cyan), and HOM monomer (light green) and dimer (dark green). All unidentified ions are shown in grey. In total, each 'waffle' contains 100 squares, therefore each square represents 1% of the total signal. The fractions of the compounds are the median of all measurements in the corresponding layers. The mixing layer was measured twice, before noon and in the afternoon. The two levels shown in the schematic are at 100 m and 700 m.

### 3.5 Summary of observations

Figure 19 shows a summary of all six measurement days at different times, and separated into the different layers and ion compositions. The figure depicts the fraction of ions containing nitric acid, sulfuric acid, MSA and $SO_5^-$, iodic acid, carboxylic acids, and HOM monomers and dimers in relation to the total measured ion signal. It is apparent that the FT did not show any ions containing HOM, but instead a large amount of nitric acid and carboxylic acids during morning hours, while during daytime, most ions contained sulfate, mostly sulfuric acid but also MSA. Iodic acid was apparent at all times of measurements in the FT. The RL showed its own ion chemistry with a variety of compounds (discussed in section 3.4.2) and it changed with each day. Therefore, in this summary figure, which is an average of all the measurements at a specific time of the day, we see HOM as well as higher amounts of iodic acid and carboxylic acids even though they might not have been present at the





same time. The SBL shows less iodic acid and more HOMs compared with the RL. Additionally, the SBL contains a large proportion of HOM dimers. Although the ions in the ML were mainly sulfuric acid and $SO_5^-$, the fraction of HOM increased in the afternoon. Overall, due to the high concentration of sulfuric acid during daytime and photo-oxidation, most of the ions measured by the APi-TOF were sulfuric acid in every layer. Nitric acid was abundant in every layer and at every time of the
day.

## 4   Conclusions

A new tailored flying APi-TOF mass spectrometer was deployed in a Cessna 172 aircraft to measure the vertical profiles of atmospheric ions above the boreal forest at the SMEAR II station, covering the stable boundary layer (SBL), residual layer (RL), mixed layer (ML) and free troposphere (FT). The measurements showed that the ion composition within the BL and FT is dis-
tinctly different, and that the ion chemistry of ML, SBL, RL, and FT is highly dependent on the time of day and air mass origin.

We detected HOM, originating from the forest VOC emissions, within the whole BL during daytime. The measured spectra in the ML were identical to a co-located reference measurement, which was performed on top of a 35-meter tower. Due to ongoing photo-oxidation during the daytime, most of the detected negative ions were the bisulfate ion and its clusters with
sulfuric acid, forming sulfuric acid dimer and trimer, and bisulfate-HOM and nitric acid-HOM clusters within the whole ML. The nocturnal RL was clearly distinguishable from the other layers by its differing ion composition and revealed its own ion chemistry with the highest day-to-day variability compared to the other layers. Depending on the air mass origin and possible decoupling of the SBL, the measured air in the RL could contain halogen compounds, MSA and carboxylic acids, and HOM. The FT showed a distinctly different ion composition than the ML, SBL, or RL. In the FT, a high abundancy of carboxylic acids
and some unknown compounds with masses between 200–400 Th were observed, probably clustered with carboxylic acids. Furthermore, we detected some very specific ion patterns on two days of measurements within the FT, which likely contained halogens (most likely chlorine), due to their apparent isotopic pattern. The daytime measurements in the FT mainly showed sulfuric acid, MSA, nitric acid and malonic acid, however we did not detect HOM.

Our observations represent only a snapshot of the ion composition in the BL and FT during springtime above the boreal
forest. More observations, especially in the RL and FT, during other times of the year would be necessary for getting a better understanding of the chemistry and behaviour of ions in those layers and, ultimately to quantify the role of ions in NPF in different parts of the global troposhere.

*Data availability.* The APi-TOF and airborne data is available on Zenodo at with the doi: 10.5281/zenodo.5741438. The SMEAR II data can be accessed from https://avaa.tdata.fi/web/smart/smear.





**Table A1.** Identified compounds of the stable boundary layer (SBL), residual layer (RL), free troposphere (FT) and mixed layer (ML) on 2 May 2017 on different heights during early morning and afternoon flight. The mass is given as the measured mass of the most abundant isotope.

| | | Early morning | | | | Afternoon | | |
| | | SBL | SBL | RL | FT | ML | ML | FT |
| Elemental composition (name) | mass (Th) | 35 m | 100 m | 700 m | 3200 m | 35 m | 1500 m | 2700 m |
|---|---|---|---|---|---|---|---|---|
| $CO_3^-$ | 59.9853 | | | | X | X | X | X |
| $NO_3^-$ (nitric acid) | 61.9884 | X | X | X | X | X | X | X |
| $Br^-$ | 78.9189 | | X | X | | | | X |
| $SO_3^-$ | 79.9574 | | | | | | | X |
| $H_2O \cdot NO_3^-$ (water · nitric acid) | 79.9989 | | X | X | | | X | |
| $CH_3SO_3^-$ (methanesulfonic acid, MSA) | 94.9808 | | X | X | | X | | X |
| $HSO_4^-$ (bisulfate ion) | 96.9601 | X | X | X | X | X | X | X |
| $C_3H_3O_4^-$ (malonic acid) | 103.0037 | X | X | X | X | X | X | X |
| $SO_5^-$ | 111.9472 | | X | X | X | X | X | X |
| $C_4H_3O_4^-$ (maleic acid) | 115.0037 | | | X | | | | X |
| $H_2CO_3 \cdot NO_3^-$ (carbonic acid · nitric acid) | 123.9888 | X | X | X | X | X | X | X |
| $HNO_3 \cdot NO_3^-$ (nitric acid dimer) | 124.984 | X | X | X | X | X | X | X |
| $I^-$ | 126.905 | X | X | X | | | X | |
| $HO_2N_2S \cdot NO_3^-$ | 154.9642 | | X | | | X | X | X |
| $CH_4SO_3 \cdot NO_3^-$ (MSA · nitric acid) | 157.9765 | | X | | | | | X |
| $H_2SO_4 \cdot NO_3^-$ (sulfuric acid · nitric acid) | 159.9557 | | X | | | X | X | X |
| $C_3H_4O_4 \cdot NO_3^-$ (malonic acid · nitric acid) | 165.9993 | X | X | X | X | X | X | X |
| $IO_3^-$ (iodic acid) | 174.8898 | X | X | X | X | X | X | X |
| $C_4H_2O_4 \cdot NO_3^-$ | 175.9837 | | X | X | X | X | X | |
| $C_4H_4O_4 \cdot NO_3^-$ (maleic acid · nitric acid) | 177.9993 | | | X | X | | X | X |
| $(HNO_3)_2 \cdot NO_3^-$ (nitric acid trimer) | 187.9797 | | | | X | | | |
| $CH_4SO_3 \cdot CH_3SO_3^-$ (MSA dimer) | 190.9690 | | | | | | | X |
| $H_2O \cdot IO_3^-$ (water · iodic acid) | 192.9003 | X | X | X | X | X | X | |
| $CH_4SO_3 \cdot HSO_4^-$ (MSA · sulfuric acid) | 192.9482 | | X | | | | X | X |
| $H_2SO_4 \cdot HSO_4^-$ (sulfuric acid dimer) | 194.9275 | | X | | | X | X | X |
| $ICl_2^-$ | 196.8427 | | X | X | | | | |
| $C_3H_7O_6 \cdot NO_3^-$ | 201.0126 | X | X | X | X | | | |
| $N_2O_5 \cdot C_3H_3O_4^-$ | 210.9844 | | | X | | | | X |
| $CH_4SO_3 \cdot C_3H_3O_5^-$ | 214.9867 | | | | | | | X |
| $C_6H_7O_3 \cdot HSO_4^-$ | 223.9996 | | | | | X | | |



|  |  | Early morning | | | | Afternoon | | |
| --- | --- | --- | --- | --- | --- | --- | --- | --- |
|  |  | SBL | SBL | RL | FT | ML | ML | FT |
| Elemental composition (name) | mass (Th) | 35 m | 100 m | 700 m | 3200 m | 35 m | 1500 m | 2700 m |
| $C_5H_6O_6 \cdot NO_3^-$ | 224.0048 |  |  | X |  |  |  |  |
| $C_5H_6O_{10}^-$ | 225.9966 | X | X |  |  | X |  |  |
| $HIO_3 \cdot NO_3^-$ (iodic acid · nitric acid) | 237.8854 | X | X | X | X | X | X | X |
| $C_5H_6O_7 \cdot NO_3^-$ | 239.9997 |  | X |  | X | X | X |  |
| $C_5H_9O_7 \cdot NO_3^-$ | 243.0232 |  | X |  |  |  | X |  |
| $C_5H_9O_8 \cdot NO_3^-$ | 259.0181 |  | X |  |  |  |  | X |
| $ICl_4^-$ | 266.7804 |  | X | X |  |  |  |  |
| $C_7H_{11}O_7 \cdot NO_3^-$ | 269.0388 |  | X |  | X |  | X |  |
| $HIO_3 \cdot HSO_4^-$ (iodic acid · sulfuric acid) | 272.8571 |  | X | X |  |  | X | X |
| $C_7H_8O_8 \cdot NO_3^-$ | 282.0103 | X |  |  |  |  |  |  |
| $C_5H_7O_9N \cdot NO_3^-$ | 287.0004 |  |  |  |  | X | X |  |
| $(H_2SO_4)_2 \cdot HSO_4^-$ (sulfuric acid trimer) | 292.8949 |  |  |  |  | X | X | X |
| $C_{13}H_{12}O_4 \cdot NO_3^-$ | 294.0619 |  | X |  |  |  |  |  |
| $C_{10}H_{15}O_{10}^-$ | 295.0671 | X | X |  |  |  |  |  |
| $C_9H_{14}O_7 \cdot NO_3^-$ | 296.0623 |  | X |  |  |  | X |  |
| $C_7H_9O_8N \cdot NO_3^-$ | 297.0212 | X | X |  |  | X | X |  |
| $C_8H_{12}O_8 \cdot NO_3^-$ | 298.0416 |  | X |  |  |  | X |  |
| $C_{11}H_9O_6 \cdot NO_3^-$ | 299.0283 |  | X |  |  | X |  |  |
| $C_7H_{11}O_9 \cdot NO_3^-$ | 301.0287 |  |  |  |  |  | X |  |
| $C_{10}H_{15}O_6N \cdot NO_3^-$ | 307.0783 |  | X | X |  |  |  |  |
| $C_{10}H_{14}O_7 \cdot NO_3^-$ | 308.0623 | X | X |  |  |  | X |  |
| $C_{10}H_{16}O_7 \cdot NO_3^-$ | 310.0780 |  | X |  |  | X | X |  |
| $C_8H_{11}O_8N \cdot NO_3^-$ | 311.0368 | X |  |  |  |  | X |  |
| $C_9H_{15}O_7N \cdot NO_3^-$ | 311.0732 | X | X |  |  |  | X |  |
| $C_9H_{14}O_8 \cdot NO_3^-$ | 312.0572 |  |  |  |  | X | X |  |
| $C_7H_{11}O_9N \cdot NO_3^-$ | 315.0317 | X | X |  |  |  |  |  |
| $C_7H_{10}O_{10} \cdot NO_3^-$ | 316.0158 | X |  |  |  |  | X |  |
| $C_{10}H_{15}O_7N \cdot NO_3^-$ | 323.0732 |  | X |  |  |  | X |  |
| $C_{10}H_{14}O_8 \cdot NO_3^-$ | 324.0572 |  |  | X |  |  | X |  |
| $C_{10}H_{15}O_8 \cdot NO_3^-$ | 325.0651 | X | X | X |  | X | X |  |
| $C_{10}H_{16}O_8 \cdot NO_3^-$ | 326.0729 |  |  |  |  | X | X |  |
| $C_9H_{15}O_8N \cdot NO_3^-$ | 327.0681 |  |  |  |  | X | X |  |
| $C_9H_{14}O_9 \cdot NO_3^-$ | 328.0521 |  | X |  |  |  |  |  |
| $C_9H_{15}O_9 \cdot NO_3^-$ | 329.0600 |  | X |  |  |  |  |  |
| $C_7H_{11}O_{10}N \cdot NO_3^-$ | 331.0267 | X |  |  |  | X | X |  |



| | | Early morning | | | | Afternoon | | |
| | | SBL | SBL | RL | FT | ML | ML | FT |
| Elemental composition (name) | mass (Th) | 35 m | 100 m | 700 m | 3200 m | 35 m | 1500 m | 2700 m |
|---|---|---|---|---|---|---|---|---|
| $C_{10}H_{15}O_8N \cdot NO_3^-$ | 339.0681 | X | X | X | | X | X | |
| $C_{10}H_{14}O_9 \cdot NO_3^-$ | 340.0521 | X | X | X | | X | X | |
| $C_{10}H_{15}O_9 \cdot NO_3^-$ | 341.0600 | X | X | | | | X | |
| $C_{10}H_{16}O_9 \cdot NO_3^-$ | 342.0678 | X | X | | | X | X | |
| $C_{10}H_{14}O_7 \cdot HSO_4^-$ | 343.0341 | X | X | | | X | | |
| $C_9H_{15}O_9N \cdot NO_3^-$ | 343.0630 | X | | | | X | | |
| $C_9H_{14}O_{10} \cdot NO_3^-$ | 344.0471 | | | | | X | | |
| $C_{10}H_{16}O_7 \cdot HSO_4^-$ | 345.0497 | X | | | | X | | |
| $HIO_3 \cdot IO_3^-$ (iodic acid dimer) | 350.7868 | | X | X | | | | |
| $C_{10}H_{15}O_9N \cdot NO_3^-$ | 355.0630 | X | X | | | | X | |
| $C_{10}H_{16}O_{10} \cdot NO_3^-$ | 358.0627 | X | X | | | X | X | |
| $C_{10}H_{14}O_8 \cdot HSO_4^-$ | 359.0290 | X | | | | X | | |
| $C_9H_{15}O_{10}N \cdot NO_3^-$ | 359.0580 | | X | | | X | X | |
| $C_{10}H_{16}O_9N_2 \cdot NO_3^-$ | 370.0739 | | X | | | | | |
| $C_{10}H_{15}O_{10}N \cdot NO_3^-$ | 371.0580 | X | X | X | | X | | |
| $C_{10}H_{14}O_{11} \cdot NO_3^-$ | 372.0420 | X | X | | | | X | |
| $C_9H_{13}O_{11}N \cdot NO_3^-$ | 373.0372 | | X | | | X | X | |
| $C_{10}H_{15}O_{11} \cdot NO_3^-$ | 373.0498 | | X | | | X | X | |
| $C_{10}H_{15}O_8N \cdot HSO_4^-$ | 374.0399 | X | | X | | X | X | |
| $C_{10}H_{16}O_{11} \cdot NO_3^-$ | 374.0576 | | | X | | X | | |
| $C_{10}H_{14}O_9 \cdot HSO_4^-$ | 375.0239 | X | | | | X | X | |
| $C_{12}H_{11}O_9N \cdot NO_3^-$ | 375.0317 | X | | | | X | | |
| $C_{10}H_{15}O_9 \cdot HSO_4^-$ | 376.0317 | X | | | | | X | |
| $C_{10}H_{15}O_{11}N \cdot NO_3^-$ | 387.0529 | X | X | | | | X | |
| $C_{10}H_{16}O_{11}N_2 \cdot NO_3^-$ | 402.0638 | | X | | | | | |
| $C_{10}H_{14}O_{11} \cdot HSO_4^-$ | 407.0137 | | | | | X | | |
| $C_{15}H_{22}O_{11} \cdot NO_3^-$ | 440.1046 | | X | | | | | |
| $C_{20}H_{32}O_{12}N_2 \cdot NO_3^-$ | 554.1839 | | X | | | | | |
| $C_{20}H_{31}O_{13}N \cdot NO_3^-$ | 555.1679 | X | | | | | | |
| $C_{20}H_{31}O_{15}N \cdot NO_3^-$ | 587.1577 | | X | | | | | |





**Table A2.** Identified compounds of free troposphere (FT) on 16 May 2017 during early morning between 04:00–05:00 EET. The FT was measured on two different height levels (1500 m and 2600 m). Compounds, which contain at least one carboxyl functional group are highlighted bold.

| Elemental composition (name) | mass (Th) |
|---|---|
| $CO_3^-$ | 59.9853 |
| $NO_3^-$ (nitric acid, NA) | 61.9884 |
| $Br^-$ | 78.9189 |
| $H_2O \cdot NO_3^-$ (water · NA) | 79.9989 |
| $CH_3SO_3^-$ (methanesulfonic acid, MSA) | 94.9808 |
| $HSO_4^-$ (bisulfate ion) | 96.9601 |
| $C_3H_3O_4^-$ (**malonic acid**) | 103.0037 |
| $SO_5^-$ | 111.9472 |
| $C_4H_3O_4^-$ (**maleic acid**) | 115.0037 |
| $CH_2O_3 \cdot NO_3^-$ (**carbonic acid** · NA) | 123.9888 |
| $HNO_3 \cdot NO_3^-$ (NA dimer) | 124.9840 |
| $C_4H_3O_5^-$ (**oxaloacetic acid**) | 130.9986 |
| $CH_4SO_3 \cdot NO_3^-$ (MSA · NA) | 157.9765 |
| $H_3H_4O_4 \cdot NO_3^-$ (**malonic acid** · NA) | 165.9993 |
| $IO_3^-$ (iodic acid, IA) | 174.8898 |
| $C_4H_2O_4 \cdot NO_3^-$ (**squaric acid** · NA) | 175.9837 |
| $CH_2O_3 \cdot C_4H_3O_4^-$ (**carbonic acid · maleic acid**) | 177.0041 |
| $C_4H_4O_4 \cdot NO_3^-$ (**maleic acid** · NA) | 177.9993 |
| $C_3H_4O_5 \cdot NO_3^-$ (**tartonic acid** · NA) | 181.9942 |
| $CH_4SO_3 \cdot HSO_4^-$ (MSA · SA) | 192.9482 |
| $H_2SO_4 \cdot HSO_4^-$ (SA dimer) | 194.9275 |
| $C_3H_4O_4 \cdot CH_3SO_3^-$ (**malonic acid** · MSA) | 198.9918 |
| $C_3H_4O_4 \cdot HSO_4^-$ (**malonic acid** · SA) | 200.9711 |
| $C_3H_4O_4 \cdot H_3H_3O_4^-$ (**malonic acid** dimer) | 207.0146 |
| $C_3H_4O_4 \cdot C_4H_3O_4^-$ (**malonic acid · maleic acid**) | 219.0146 |
| $C_4H_4O_4 \cdot C_4H_3O_4^-$ (**maleic acid** dimer) | 231.0146 |
| $HIO_3 \cdot NO_3^-$ (IA · NA) | 237.8854 |
| $N_2O_3 \cdot IO_3^-$ | 250.8807 |
| $C_6H_8O_7 \cdot NO_3^-$ (**citric acid** · NA) | 254.0154 |
| $HIO_3 \cdot HSO_4^-$ (IA · SA) | 272.8571 |
| $(H_2SO_4)2 \cdot HSO_4^-$ (SA trimer) | 292.8949 |



*Author contributions.* LJB, JD, HJ, FK, DRW developed the tailored APi-TOF mass spectrometer setup. LJB, JD, HJ, JL, MK, and TP designed the field campaign. LJB, HJ, SS, IP, LLJQ, JL, KL, AM, AF, PP, DW, LD performed the field studies. LJB, SS, HJ and XCH performed the APi-TOF data analysis. AM analysed the Lidar data. LD calculated the sulfuric acid proxy and formation rates. LJB, JD, VMK, XCH, SS and TP wrote the manuscript. All authors contributed to discussions and manuscript commenting.

*Competing interests.* The authors declare no conflict of interest.

*Acknowledgements.* We acknowledge the following projects: ACCC Flagship funded by the Academy of Finland grant number 337549, Academy professorship funded by the Academy of Finland (grant no. 302958), Academy of Finland projects no. 1325656, 316114, 325647, 296628, 317380, and 320094. Russian Mega Grant project "Megapolis - heat and pollution island: interdisciplinary hydroclimatic, geochemical and ecological analysis" application reference 2020-220-08-5835, "Quantifying carbon sink, CarbonSink+ and their interaction with air quality" INAR project funded by Jane and Aatos Erkko Foundation, European Research Council (ERC) project ATM-GTP Contract No.
742206 and programme GASPARCON, grant agreement No. 714621, European Regional Development Fund (project MOBTT42), Estonian Research Council (project PRG714). Technical and scientific staff in Hyytiälä SMEAR II stations is acknowledged. The Doppler lidar data used in this study was provided from the Finnish Meteorological Institute Doppler lidar network managed by Ewan O'Connor. The authors gratefully acknowledge the NOAA Air Resources Laboratory (ARL) for the provision of the HYSPLIT transport and dispersion model and/or READY website (https://www.ready.noaa.gov) used in this publication. We thank the tofTools team for providing the tools for
the mass spectrometry analysis. We thank Alexey Voronov for providing the code for open streetmap (https://www.github.com/alexvoronov/ plot_openstreetmap). We thank Federico Bianchi, Olga Garmash, Matthias Maasch, Dominik Stolzenburg and Matthew Boyer for helpful discussions. We thank Derek Ho for the language editing of the manuscript. We thank Erkki Järvinen and the pilots at Airspark Oy for operating the aircrafts for our measurements and the hospitality and support throughout the campaign.



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
