# Peer review of "Diurnal evolution of negative atmospheric ions above the boreal forest: From ground level to the free troposphere"

_Atmospheric Chemistry and Physics, 2021_

## Author Response (AR1)

On behalf of the co-authors, I would like to thank the reviewer for the valuable comments and input which helped to improve the quality of the paper. Our responses to the comments are written in green font.

RC1:

Summary:

The authors measured negative atmospheric ions above the SMEAR II boreal forest site from a novel aircraft platform. Level flight legs were performed in the stable boundary layer, residual layer, mixed layer, and free troposphere at various times of day across several weeks. The measurements illustrate the evolution of negative ions in the atmosphere as the boundary layer rises. Ion composition is variable between different layers on each day, and variable within each layer across different days due to changes in airmass history. I found this paper to be very clearly written and depicted, and it is a useful contribution to the knowledge of atmospheric ions and their impacts on new particle formation. I have just a few minor comments below, and recommend publication in ACP.

Comments:

Line 176: Can you provide more details here for readers who may be unfamiliar with sampling with an APi-TOF, e.g., are ion transmission rates of ~1% normal? How do your transmissions rates compare with previous field measurements of ions, or with the ground measurements at the SMEAR II site?

AC: The measured transmission of the APi-TOF with 1% is a common characteristic of this instrument. It has also been shown in publications, e.g., Heinritzi et al. (2016) and Junninen et al. (2010). The characterisation of the transmission of the APi-TOF is usually done in the laboratory due to the rather bulky setup. So, therefore we have done the calibration in the lab before the campaign (as mentioned in the text). The inter-comparison of the flying APi-TOF and the stationary APi-TOF showed that the transmission of both of these instruments are around 1%.

In order to give more insight to the reader regarding the transmission, we added the following sentence to the manuscript (italic):

A transmission characterisation of the full setup (including the sliding inlet-system for the APi-TOF) was performed in the laboratory before the campaign. The transmission of ions between 100 Th and 400 Th was in the range 1–2%, and for ions >500 Th the transmission was below 0.5% (Fig. S3). *Those transmission values match with the characterization of other APi-TOFs, e.g., Heinritzi et al. (2016) and Junninen et al. (2010).*

Line 220: When the flight setup was operated in the tower, did it include the same sliding inlet system to see if the inlet contributed to any differences between instruments?

AC: Yes, we kept the inlet system from the Cessna setup attached to the APi-TOF, and also operated it with the same vacuum pump, flow rates and instrument settings, including the regular cycle of x-ray in the inlet, in order to have the same measurement circumstances as during the flight.

Line 374: You are saying the FT air had marine origin in both morning and afternoon flights on 2 May 2017. You justify this as the reason for measuring halogenated compounds in the morning and MSA in the afternoon. Do you have any thoughts on why the halogenated compounds were not present in the afternoon and MSA was not present in the morning? Is this a result of charge transfer, as you discuss later on (e.g. line 438), or something else?

AC: The charge transfer to the sulfur containing compounds is certainly one of the main reasons why we primarily observe sulfuric acid and MSA rather than halogens in the afternoon.

The origin of gas-phase MSA is the oxidation of DMS via OH, so MSA is formed during daytime. The lifetime of DMS is 1-2 days (Boucher et al., 2003; Chen et al., 2018), and according to the 72 h back trajectories in the supplementary Figure S15 (c) (see below), the air mass was above the sea > 12 hours before arriving above SMEAR II station at 13 UTC (i.e. 15 EET). The back trajectories also show that the source changes from the West coast of Norway (Norwegian Sea) in the early morning to the North coast (Barents Sea) in the afternoon. Thus, it is more likely that we see MSA only in the afternoon due to the air mass change. According to Lana et al. (2010), the DMS production in May is higher in the Barents Sea than in the Norwegian Sea, and it is likely that the MSA concentration within that air mass was higher than in the air mass of the morning hours.

[Figure]

Line 478: I wonder if you find any correlation between ground level monoterpene concentrations and/or ozone concentrations with HOM in the SBL. Maybe there is a way to explain some of the slight variations in HOM across different days. Were there monoterpene measurements at SMEAR II during these times?

AC: Indeed, the ozone and / or monoterpene concentration were relatively high during the days with high amounts of HOM in the SBL. Below we show a figure with the time series of monoterpenes (MT), $O_3$, $NO_x$ and the temperature (at the 4.2-m height). Unfortunately, there is a data gap in the MT concentration on 5 May 2017 during the morning hours, however the concentration in the previous evening reached 0.8 ppb at 19:29 local time before the data gap appeared. Possibly, the concentration was still rather high on 5 May morning.

The table below shows the average values of $O_3$ and MT during the measurements. The maximum of $O_3$ concentration was reached on 5 May and the maximum of MT concentration on 16 May 2017, when a high amount of especially HOM dimer were observed. The abundance of $O_3$ and MT could explain the high amount of detected HOM (dimers).

Table 1: Median of Ozone and monoterpene (MT) concentration at 4.2 m height during early morning hours when the measurements were taken in the SBL. On 5 May 2017, no MT data was available, therefore the last measured value from the day before is shown in the table.

|  | $O_3$ (ppb) | MT (ppb) |
|---|---|---|
| 2 May (04:00 - 04:30) | 39 | 0.08 |
| 3 May (04:00 - 04:30) | 34 | 0.16 |
| 4 May (04:00 - 04:30) | 39 | 0.09 |
| 5 May (04:00 - 04:30) | **40** | (4 May 19:29: MT = 0.8 ppb) |
| 12 May (03:30 - 04:00) | 28 | 0.18 |
| 16 May (03:30 - 04:00) | 32 | **0.3** |

[Figure]

Figure 1: Monoterpene (MT) (green), Ozone (grey) and NOx concentration (blue) given in ppb, as well as temperature in °C (red) at SMEAR II station from 2 May to 16 May 2017 at 4.2 m height above ground level.

[Figure]

Figure 12 from the manuscript: Mass defect of the detected ions in the SBL at 100 m height. Each measurement day is shown separately. Nitric acid containing ions are coloured dark blue, and ions and clusters with sulfuric acid, MSA, or both and the the $SO_5^-$ ion are indicated in red. The compounds containing carboxyl functional groups are shown in cyan, and iodic acid and iodine-containing compounds are shown in yellow. The size of the dots corresponds to the measured signal (ions s$^{-1}$).

Fig S2: Can you colour by altitude? That would help make the plot clearer I think.

[Figure]

***Figure S2*** *Time series of pressures inside the APi-TOF during the measurement campaign. (a) Pressure of the Small Segmented Quadrupole (SSQ) chamber inside the APi-TOF as well as **(b)** the Big Segmented Quadrupole (BSQ) during all measurement flights. The pressure of the SSQ was kept stable at 2 mbar on 4 out of 6 measurement days (4, 5, 12 and 16 May 2017). The BSQ pressure dropped with the ambient pressure on each flight, but did not reach values below 0.025 mbar during flights when the SSQ was at 2 mbar.* *The depicted pressures are coloured by height in m AGL.*

AC: Thank you for the suggestion. We added the color for the height and also removed some data between the actual flight measurements to clean up the figure. Note that the pressure of the SSQ chamber was kept constant at 2 mbar in the last 4 out of 6 measurement days, the colors overlap and the values above 500 m are barely visible.

General comment: This paper focuses mainly on presenting the data, and it would be even more useful if you could expand upon what these results mean for atmospheric processes that depend on ions. For instance, what does the variable distribution of negative ions mean for the likelihood of NPF in each different layer of the atmosphere? Is NPF favored by the ions found in the boundary layer and mixed layer, or disfavored by the lack of HOM in the free troposphere? Please expand somewhere, perhaps in the conclusions section, even if it is to clarify that not enough is known yet about what role each subclass of ions ultimately plays in NPF.

AC: Considering that we only show observations of about 40 hours of measurement (i.e. 16 flights performed in 5 days) during springtime, we were rather careful with drawing too big conclusions, but we added the following paragraph to the conclusion section (green font).

*From the conclusion:*

*A new tailored flying APi-TOF mass spectrometer was deployed in a Cessna 172 aircraft to measure the vertical profiles of atmospheric ions above the boreal forest at the SMEAR II station, covering the stable boundary layer (SBL), residual layer (RL), mixed layer (ML) and free troposphere (FT). The measurements showed that the ion composition within the BL and FT is distinctly different, and that the ion chemistry of ML, SBL, RL, and FT is highly dependent on the time of day and air mass origin.*

*We detected HOM, originating from the forest VOC emissions, within the whole BL during daytime. The measured spectra in the ML were identical to a co-located reference measurement, which was performed on top of a 35-meter tower. Due to ongoing photo-oxidation during the daytime, most of the detected negative ions were the bisulfate ion and its clusters with sulfuric acid, forming sulfuric acid dimer and trimer, and bisulfate-HOM and nitric acid-HOM clusters within the whole ML. The nocturnal RL was clearly distinguishable from the other layers by its differing ion composition and revealed its own ion chemistry with the highest day-to-day variability compared to the other layers. Depending on the air mass origin and possible decoupling of the SBL, the measured air in the RL could contain halogen compounds, MSA and carboxylic acids, and HOM. The FT showed a distinctly different ion composition than the ML, SBL or RL. In the FT, a high abundance of carboxylic acids and some unknown compounds with masses in the range 200–400 Th were observed, probably clustered with carboxylic acids. Furthermore, we detected some very specific ion patterns on two days of measurements within the FT, which likely contained halogens (most likely chlorine), due to their apparent isotopic pattern. The daytime measurements in the FT mainly showed sulfuric acid, MSA, nitric acid and malonic acid, however we did not detect HOM.*

As we summarised in the introduction, ion-induced nucleation is the main NPF pathway in several environments, especially under conditions when abundances of nucleating precursor gases are relatively low. At high enough precursor concentrations, ion-induced nucleation becomes less relevant for initiating atmospheric NPF. The compounds we observed in the various layers favour NPF, and even though our measurements focused on ions rather than neutral vapours, the ion chemistry sheds light on the available precursor gases. From our observations we could conclude that in the FT above the boreal forest, NPF could be initiated via ion-induced nucleation of inorganic compounds, such as sulfuric acid and iodic acid, also under the aspect that lower temperatures favour the nucleation (e.g. He et al., 2021; Duplissy et al., 2016). Within the boundary layer, however, it is more likely that a mixture of various compounds initiates the nucleation, especially as HOM have been observed within the whole ML. Furthermore, NPF has been observed at Jungfraujoch (Bianchi et al., 2016) and the Himalayan mountains (Bianchi et al., 2021) when organic compounds were transported upwards and mixed into the free troposphere. Also Rose et al. (2015) observed NPF at Puy de Dôme, at the interface between the boundary layer and FT, albeit nucleation there was driven by neutral compounds rather than via ion-induced pathways. Still, this could

indicate that a possible mixing of the organics in the mixing layer with the inorganics in the FT would provide good conditions for nucleation, e.g. during roll vortex events (Lampilahti et al., 2020). Thus, it could be speculated that the multi-component nucleation as proposed by Lehtipalo et al. (2018) is driving the nucleation. Moreover, above we discussed the nucleation mechanisms and nucleating precursor vapours, while it should not be forgotten that for NPF as a whole, also vapours that further grow small clusters into larger sizes are highly relevant.

*Our observations represent only a snapshot of the ion composition in the BL and FT during springtime above the boreal forest. More observations, especially in the RL and FT, during other times of the year would be necessary for getting a better understanding of the chemistry and behaviour of ions in those layers and, ultimately to quantify the role of ions in NPF in different parts of the global troposphere.*

References:

Boucher, O., Moulin, C., Belviso, S., Aumont, O., Bopp, L., Cosme, E., von Kuhlmann, R., Lawrence, M. G., Pham, M., Reddy, M. S., Sciare, J., and Venkataraman, C.: DMS atmospheric concentrations and sulphate aerosol indirect radiative forcing: a sensitivity study to the DMS source representation and oxidation, Atmos. Chem. Phys., 3, 49–65, https://doi.org/10.5194/acp-3-49-2003, 2003.

Chen, Q., Sherwen, T., Evans, M., and Alexander, B.: DMS oxidation and sulfur aerosol formation in the marine troposphere: a focus on reactive halogen and multiphase chemistry, Atmos. Chem. Phys., 18, 13617–13637, https://doi.org/10.5194/acp-18-13617-2018, 2018.

Heinritzi, M., Simon, M., Steiner, G., Wagner, A. C., Kürten, A., Hansel, A., and Curtius, J.: Characterization of the mass-dependent transmission efficiency of a CIMS, Atmos. Meas. Tech., 9, 1449–1460, https://doi.org/10.5194/amt-9-1449-2016, 2016.

Junninen, H., Ehn, M., Petäjä, T., Luosujärvi, L., Kotiaho, T., Kostiainen, R., Rohner, U., Gonin, M., Fuhrer, K., Kulmala, M., and Worsnop, D. R.: A high-resolution mass spectrometer to measure atmospheric ion composition, Atmos. Meas. Tech., 3, 1039–1053, https://doi.org/10.5194/amt-3-1039-2010, 2010.

Lana, A., et al: An updated climatology of surface dimethlysulfide concentrations and emission fluxes in the global ocean, *Global Biogeochem. Cycles*, 25, GB1004, doi:10.1029/2010GB003850, 2011.

**From ACP editor on February 20th (Roya Bahreini):**

*Dear All,*

*Given the unique set of measurements provided here, I would like to consider this manuscript for publication as a Research article (and not a Measurement Report) if you can provide more context on atmospheric implications of the observed trends. Please let me know if you have any questions.*

—-------------------------------------------------------------------------------------------------------

On behalf of the co-authors, I would like to thank the reviewer for the valuable comments and inputs which helped to improve the quality of the paper. Our responses to the comments are written in green font.

**RC2: Summary and general comment:**

This study by Beck et al. characterizes the atmospheric negative ion composition of various tropospheric layers (from 100m to 3200m) above the SMEAR II research station in the Finnish boreal forest. By deploying an API-TOF mass spectrometer in a Cessna 172 aircraft, the authors resolved the vertical distribution and diurnal variation of naturally charged molecular ions and ion-clusters from the stable boundary layer up to the free troposphere. While the ion composition of the stable boundary layer coincides with the simultaneous measurements placed at the field station on top of a 35m tower above the forest canopy, the other layers showed a clearly distinguishable ion composition. In the residual and mixing layers, the ion composition is strongly dependent on the origin of air masses, the turbulent mixing, and the photochemistry throughout the day. Most of the detected ions during daytime are comprised of sulfur-containing compounds, while the nocturnal ion compositions are more diverse. The detection of various halogens and carboxylic acids as well as the absence of HOMs in the free troposphere indicates that this layer is mainly influenced by long-term transported air mass.

I agree with the general comment by reviewer 1. The authors realized a difficult technical measurement and showed through comprehensive investigations the composition and chemical processes of negative ions in various tropospheric layers and their diurnal evolution. The methods are extensively illustrated and clearly written, enabling a good evaluation of the results by the scientific community, and promoting a continuation of the study in this field of research. However, the manuscript lacks a detailed discussion and developed conclusion of the atmospheric implication to obtain a comprehensive understanding of ion chemistry and its role for NPF in the various tropospheric layers. I would recommend the manuscript for publication in ACP as a measurement report after the following minor comments have been addressed.

AC: Considering that we only show observations of about 40 hours of measurement (i.e. 16 flights performed in 5 days) during springtime, we were rather careful with drawing too big conclusions, but we added the following paragraph to the conclusion section (green font).

[...] As we summarised in the introduction, ion-induced nucleation is the main NPF pathway in several environments, especially under conditions when abundances of nucleating precursor gases are relatively low. At high enough precursor concentrations, ion-induced nucleation becomes less relevant for initiating atmospheric NPF. The compounds we observed in the various layers favour NPF, and even though our measurements focused on ions rather than neutral vapours, the ion chemistry sheds light on the available precursor gases. From our observations we could conclude that in the FT above the boreal forest, NPF could be initiated via ion-induced nucleation of inorganic compounds, such as sulfuric acid and iodic acid, also under the aspect that lower temperatures favour the nucleation (e.g. He et al., 2021; Duplissy et al., 2016). Within the boundary layer, however, it is more likely that a mixture of various compounds initiates the nucleation, especially as HOM have been observed within the whole ML. Furthermore, NPF has been observed at Jungfraujoch (Bianchi et al., 2016) and the Himalayan mountains (Bianchi et al., 2021) when organic compounds were transported upwards and mixed into the free troposphere. Also Rose et al. (2015) observed NPF at Puy de Dôme, at the interface between the boundary layer and FT, albeit nucleation there was driven by neutral compounds rather than via ion-induced pathways. Still, this could indicate that a possible mixing of the organics in the mixing layer with the inorganics in the FT would provide good conditions for nucleation, e.g. during roll vortex events (Lampilahti et al., 2020). Thus, it could be speculated that the multi-component nucleation as proposed by Lehtipalo et al. (2018) is driving the nucleation. Moreover, above we discussed the nucleation mechanisms and nucleating precursor vapours, while it should not be forgotten that for NPF as a whole, also vapours that further grow small clusters into larger sizes are highly relevant. [...]

*Minor comments:*

Line 35ff.: Please also consider the earlier work of Yu, F., and R. P. Turco (2001), From molecular clusters to nanoparticles: Role of ambient ionization in tropospheric aerosol formation, J. Geophys. Res., 106, 4797–4814, 10.1029/2000JD900539.

AC: The reference was added.

Line 49: Please also consider the earlier work of Lovejoy, E. R., Curtius, J., Froyd, K. D. (2004), 'Atmospheric ion-induced nucleation of sulphuric acid and water', J. Geophys. Res. 109, 10.1029/2003JD004460.

AC: The reference was added.

Line 153: I wonder if the active ionization by the x-ray source during the flights might provide some helpful information (even of its short period) to support some conclusions.

AC: When we deployed the flying APi-TOF setup in the tower next to the (CI-)APi-TOF, we tried to find various compounds in the x-ray data. An example is shown in the figure below, where we tried to find the identified HOM compounds from CI-APi-TOF data in the x-ray data. As the sample air of the flying APi-TOF was exposed to direct x-ray, we assumed the compounds could be visible either deprotonated, as $NO_3^-$ clusters or as clusters with $O_2^-$, which is produced due to the x-ray in the inlet. However, as the example in the figure shows, the deprotonated and $O_2^-$ compounds do not agree with the CI-APi-TOF HOM signal. Although the $NO_3^-$ clusters seem to agree with the detected HOM compound to some degree, there are some unpredictable errors during daytime. Thus, the peak identification is rather complex because several compounds are be included within one peak. We observed the same behaviour with other compounds. We might have been able to distinguish the compounds if our instrument would have a higher resolution.

[Figure]

Red: $C_{10}H_{14}O_{11} \cdot NO_3-$ measured with CI-APi-TOF

Blue: testing if the HOM compound ($C_{10}H_{15}O_{11}$) as seen with CI-APi-TOF can be found in the direct x-ray data, either:
(A) deprotonated,
(B) clustered with $NO_3-$ or
(C) with $O_2-$

The values on the vertical axes are from the raw signal (ions s$^{-1}$)

Figure 1: Comparison between detected HOM compound with CI-APi-TOF (red line), using nitrate as a reagent ion, and the x-ray signal from the flying APi-TOF. We searched for various options of ionized HOM (blue line), such as the deprotonated one (A), the HOM clustered with $NO_3^-$ (B) and clustered with $O_2^-$ (C).

Line 155: Did you observe a significant variation of the total ion count (TIC or ions per second, including transmission correction) during the flights in the different tropospheric layers or times of the day?

Table 1: Average total ion count (TIC) per layer and per time of day from dawn to afternoon. Since the RL and SBL are mixed into the ML during daytime, the layers were kept in the same row.

| Total ion count (TIC) per layer (rows) per time of day (columns) | Dawn | Morning | Afternoon |
|---|---|---|---|
| FT | 14 | 19 | 16 |
| RL (dawn)
ML at 700 m (morning & afternoon) | 9 | 11 | 7 |
| SBL (dawn)
ML at 100 m (morning & afternoon) | 8 | 12 | 7 |

AC: The table above shows the average TIC per layer and per time of day. When interpreting the numbers, however, it should be kept in mind that some TIC numbers contain the average of 2 measurement days (FT afternoon and ML 700 m afternoon) while other layers contain the average of 3-5 measurements.

The FT shows a higher ion count than the lower layers. This would be reasonable, since the FT has a lower condensation sink and the ion production rate might be increased due to galactic cosmic radiation. The ML during morning hours has a higher ion count than during afternoon, and a possible reason might be the increase in the condensation sink, as each measurement day was a NPF day, and NPF occurred around noon. However, more data would be necessary to draw any larger conclusions here.

Line 163ff.: You may add the TIC to Figure S2 if there is a variation.

AC: The TIC was added in Figure S2, see below. Note that we also implemented the suggestion by RC1 and coloured the data points by height.

[Figure]

**Figure S2** Time series of pressures inside the APi-TOF during the measurement campaign. (a) Pressure of the Small Segmented Quadrupole (SSQ) chamber inside the APi-TOF as well as **(b)** the Big Segmented Quadrupole (BSQ) during all measurement flights. The pressure of the SSQ was kept stable at 2 mbar on 4 out of 6 measurement days (4, 5, 12 and 16 May 2017). The BSQ pressure dropped with the ambient pressure on each flight but did not reach values below 0.025 mbar during flights when the SSQ was at 2 mbar. The depicted pressures are coloured by height in m AGL. Panel (c) depicts the total ion count (ions s$^{-1}$), also coloured by height.

Line 164ff.: Can you estimate qualitatively how strong the measurements can be influenced by the pressure changes (e.g., transmission and ion-cluster stability).

AC: The instrument was equipped with a pressure adapting valve for the first quadrupole chamber. This kept the pressure after the pinhole constant at 2 mbar on all altitudes. The device did not function fully on the first two measurement days (2nd and 3rd May 2017, see figure above). In theory, that should also keep the pressures in the subsequent lower-pressure chambers constant, and hence ion transmission or fragmentation would not be affected. However, in retrospect we noticed that pressure of other chambers of the instrument did change with altitude. The exact reason for this is not clear, but most likely this happened due to the change in backpressure of the pre-vacuum pump and turbo pump, thus causing the pressures within the APi-TOF to drop with higher altitudes. Even though the pressure of the first chamber remained constant. The majority of fragmentation happens in the first chamber and also at the transmission to the second chamber.

In order to understand the effect of the pressure changes on transmission and fragmentation, one would need to conduct proper chamber measurements with constant and quantified ion-cluster concentrations, where the pressures can be altered, similar as done in the study by Alfaouri et al. (2022) (https://doi.org/10.5194/amt-15-11-2022). In their study, however, they focused on the voltages within the quadrupoles, not on the pressure difference itself.

Figure 5: Please mention the black dotted line and its meaning in the figure caption.

AC: We added as follows:

*"[...] The measurements were used to estimate the height and vertical extent of the different layers of the boundary layer, such as stable layer, residual layer, mixed layer and free troposphere. The red line on the time axis indicates the airborne measurement periods.* The black dotted line indicates the height of the ML. *[...]"*

Line 371ff: Similar to reviewer 1; you showed that during daytime MSA is present in the FT, while iodic acid and halogenated compounds are almost absent. On the basis that the measured air mass had a marine origin (as shown in Fig. S15), can you elaborate a bit more on your conclusion? Typically, you would expect all species from marine sources. Can MSA also be formed by a different source, e.g., degassing of MSA from dehydrated aerosol (Zhang et al. (2014); doi: 10.1002/2014GL060934).

AC: Certainly, there is a possibility that the detected MSA might be originating from dehydrated aerosol in the FT, as proposed by Zhang et al. (2014). However, we cannot verify nor exclude this possibility with our measurements. For the sake of completeness, we thus added the reference to the manuscript as follows:

[...] In the FT, most of the detected ions were sulfuric acid monomer, dimer and trimer as well as $SO_5^-$ and nitric acid. Unlike the ML, the FT also contained MSA monomer and dimer, whereas HOM were not abundant at all. According to the HYSPLIT back trajectories (Fig. S15), the air mass measured in the afternoon in the FT had a marine origin, explaining the presence of MSA.

*Another possible source of MSA in the FT might be the degassing from dehydrated aerosol, as observed and modelled by Zhang et al. (2014). [...]*

AC: The reason why iodic acid and other halogenated compounds are rather absent during the daytime might be due to the fact that sulfuric acid and MSA have a lower proton affinity and are more likely to be detected with our ambient ion measurements.

Figure 7: What element composition corresponds to the strong iodine peak in between the sulfur-containing molecules close to 200Th? Please mark/add this one also in table A1.

AC: The compound was falsely coloured, thank you for noticing! In the other layers the compound at 193 Th is $H_2O \cdot IO_3^-$, however in the FT the compound at 193 is $CH_4SO_3 \cdot HSO_4^-$ (MSA and sulfuric acid). The compounds are already correctly mentioned in table A1, and the figure was updated with the correct colouring:

[Figure]

Figure 2: Spectra from the original figure 7 in the manuscript, but with corrected peak colouring for the compound at 193 Th.

Line 409ff.: Are any ammonia-containing clusters measured at the ground station during this time to support your conclusion for this case study?

AC: We did not detect any sulfuric acid – ammonia clusters in our spectrum in the tower measurements. But it should be mentioned that the data from the instrument in the tower was rather noisy, which increased the limit of detection. Thus, the clusters might have been abundant but below the limit of detection and therefore not visible with our instrument.

Line 436ff.: The concentration of HOM dimers and iodic acid could also be reduced due to NPF or atmospheric chemical processes. In my opinion, the statement "their charges being transferred to SA" is not fully valid here. Please elaborate a bit more on this conclusion.

[Figure]

Figure 3: Time series of sulfuric acid (SA, red), iodic acid (IA, yellow), HOM monomer (light green, solid) and HOM dimer (dark green, dashed) measured with nitrate CI-APi-TOF on the tower at SMEAR II station. The black windows on 16 May 2017 show the measurement time windows of ascent and descent at 100 m height.

AC: The figure above shows the time series of sulfuric acid, iodic acid, HOM monomers and dimers from 14 May to 17 May 2017, measured by nitrate-CI-APi-TOF on the tower at SMEAR II station. The behaviour of the iodic acid concentration agrees with previous observations, e.g., He et al. (2021) (see supplementary material, Fig. S9, file:///Users/lisabeck/Downloads/abe0298_he_sm-1.pdf). Since the iodic acid concentration is increasing between the first and the second measurement (see black boxes) from $3 \cdot 10^5$ to $8 \cdot 10^5$ molecules cm$^{-3}$, it is more likely that iodic acid is still abundant but not visible in our APi-TOF measurements due to the charge transfer to sulfuric acid, the concentration of which increases from $6 \cdot 10^5$ to $4 \cdot 10^6$ molecules cm$^{-3}$.

The HOM dimer concentration is decreasing between the two measurements. The decrease during morning hours agrees with previous observations, e.g. Bianchi et al. (2017) (https://doi.org/10.5194/acp-17-13819-2017). Furthermore, our APi-TOF spectra suggest a compositional change in the ions from HOM·NO$_3^-$ to more HOM·HSO$_4^-$ ions, as also observed during morning hours by Yan et al. (2016) (doi:10.5194/acp-16-12715-2016), thus a charge scavenging by sulfuric acid. This process would then mask potential reductions of HOM ions due to NPF.

Figure 6 + 10: You may want to add the estimated sulfuric acid concentration also in these figures as shown in Figure 7b.

AC: We decided not to include the estimated sulfuric acid concentration in these figures, since the method works well during daytime, when the sulfuric acid clusters are abundant, but not so well during night time (see Beck et al., 2022), when the sulfuric acid cluster concentration is rather low, as can be determined from the spectra in the plots as well. Thus, not to mislead the reader with potentially wrong concentrations, we rather left the profiles out in the early morning hour observations.

Figure 11+12+13+14+16+18: Why are the detected HOMs (shown in the figures before and table A1) now combined with all (unidentified ion compounds)?

AC: Most of the HOM compounds are not single peaks, i.e. several compounds (and their isotopes) have a similar mass and thus one peak in the spectra contains several HOM compounds (and / or their isotopes). We used a simplified classification of HOM in the figures 6, 7 and 10, and coloured all peaks > 300 Th green, to indicate that the majority of those compounds are HOM. The resolution of those figures is done with unit-mass resolution, i.e. all compounds in a mass range of e.g. 300 Th ± 0.5 Th are included in one depicted peak.

In the figures 12, 14, 16 and 18 we show the high-resolution of the signal. Here, we did not use a colour code, because there is only a small fraction of identified HOM compared to many unidentified ones. Furthermore, some of the high-resolution peaks might not be HOM and we wanted to avoid false colouring.

In the figures 11 and 13 we are showing a "zoom-in" on the signal and the HOM pattern are more clearly visible. Here, we did not colour all compounds green due to the same reason as above, that not all compounds are necessarily HOM. However, to not confuse the reader with our decision of colouring, we added to the figure caption of those two figures as follows:

*Figure 11. Measurements from 100 m above the ground during early morning between 03:45 and 04:15 EET. Ion spectra measured with APi-TOF (a.1–f.1). The mass axis is split at 300 Th such that the limits of the vertical axis could be decreased to make the ions >300 Th more visible. The composition of identified ions with a high signal are noted in the figure. Please note that unlike in Fig. 6, 7 and 10, the signal > 300 Th is not coloured green. The majority of the compounds is HOM dominated, but also other ions are abundant and to avoid possible false colour coding in this higher resolution figure, the colour is kept neutral. […]*

AC: Finally, to distinguish between the HOM monomer and dimer in the overview figure 19, we used two shades of green so that the evaluation of monomer and dimer are more visible.

*Technical comments:*

Line 254: better use "sources of uncertainty" instead of "source of error".

AC: Changed according to the suggestion.

Line 276: better use "upward trend" instead of "buoyancy".

AC: Changed according to the suggestion.

Line 385: The estimated sulfuric acid concentration [...] varied from 0.3 to $1 \cdot 10_6$ cm-3 and [...].

AC: Thank you for noticing, this was corrected.

Line 402: reference should be Fig. S15, not S6.

AC: The reference was corrected.

Line 505: You may focus the reader on the trajectory you referring to (e.g., Fig. S14 blue trace).

AC: The suggestion was implemented.

Reference to Fig. S17 is missing.

AC: We added the reference to Fig. S17 in chapter 3.1. (Campaign overview) as follows:

[...] The temperature at 2 m height varied between -5 and +18°C (Fig. 3c) and between -4 and +16°C above the canopy (33.6 – 125 m, Fig. S17). [...]

Alfaouri, D., Passananti, M., Zanca, T., Ahonen, L., Kangasluoma, J., Kubecka, J., Myllys, N., Vehkamäki, H., A study on the fragmentation of sulfuric acid and dimethylamine clusters inside an atmospheric pressure interface time-of-flight mass spectrometer, Atmospheric Measurement Techniques, 15, 11--19, 10.5194/amt-15-11-2022, 2022

Beck, L. J., Schobesberger, S., Sipilä, M., Kerminen, V.-M., and Kulmala, M.: Estimation of sulfuric acid concentration using ambient ion composition and concentration data obtained with atmospheric pressure interface time-of-flight ion mass spectrometer, Atmospheric Measurement Techniques, 15, 1957–1965, https://doi.org/10.5194/amt-15-1957-2022, 2022

Bianchi, F., Garmash, O., He, X., Yan, C., Iyer, S., Rosendahl, I., Xu, Z., Rissanen, M. P., Riva, M., Taipale, R., Sarnela, N., Petäjä, T., Worsnop, D. R., Kulmala, M., Ehn, M., and Junninen, H.: The role of highly oxygenated molecules (HOMs) in determining the composition of ambient ions in the boreal forest, Atmospheric Chemistry and Physics, 17, 13 819–13 831, https://doi.org/10.5194/acp-17-13819- 2017, 2017.

He, X.-C., Tham, Y. J., Dada, L., Wang, M., Finkenzeller, H., Stolzenburg, D., Iyer, S., Simon, M., Kürten, A., Shen, J., Rörup, B., Rissanen, M., Schobesberger, S., Baalbaki, R., Wang, D. S., Koenig, T. K., Jokinen, T., Sarnela, N., Beck, L. J., Almeida, J., Amanatidis, S., Amorim, A., Ataei, F., Baccarini, A., Bertozzi, B., Bianchi, F., Brilke, S., Caudillo, L., Chen, D., Chiu, R., Chu, B., Dias, A., Ding, A., Dommen, J., Duplissy, J., El Haddad, I., Gonzalez Carracedo, L., Granzin, M., Hansel, A., Heinritzi, M., Hofbauer, V., Junninen, H., Kangasluoma, J., Kemppainen, D., Kim, C., Kong, W., Krechmer, J. E., Kvashin, A., Laitinen, T., Lamkaddam, H., Lee, C. P., Lehtipalo, K., Leiminger, M., Li, Z., Makhmutov, V., Manninen, H. E., Marie, G., Marten, R., Mathot, S., Mauldin, R. L., Mentler, B., Möhler, O., Müller, T., Nie, W., Onnela, A., Petäjä, T., Pfeifer, J., Philippov, M., Ranjithkumar, A., Saiz-Lopez, A., Salma, I., Scholz, W., Schuchmann, S., Schulze, B., Steiner, G., Stozhkov, Y., Tauber, C., Tomé, A., Thakur, R. C., Väisänen, O., Vazquez-Pufleau, M., Wagner, A. C., Wang, Y., Weber, S. K., Winkler, P. M., Wu, Y., Xiao, M., Yan, C., Ye, Q., Ylisirniö, A., Zauner-Wieczorek, M., Zha, Q., Zhou, P., Flagan, R. C., Curtius, J., Baltensperger, U., Kulmala, M., Kerminen, V.-M., Kurtén, T., Donahue, N. M., Volkamer, R., Kirkby, J., Worsnop, D. R., and Sipilä, M.: Role of iodine oxoacids in atmospheric aerosol nucleation, Science, 371, 589–595, https://doi.org/10.1126/science.abe0298, 2021.

Yan, C., Dada, L., Rose, C., Jokinen, T., Nie, W., Schobesberger, S., Junninen, H., Lehtipalo, K., Sarnela, N., Makkonen, U., Garmash, O., Wang, Y., Zha, Q., Paasonen, P., Bianchi, F., Sipilä, M., Ehn, M., Petäjä, T., Kerminen, V.-M., Worsnop, D. R., and Kulmala, M.: The role of H2SO4-NH3 anion clusters in ion-induced aerosol nucleation mechanisms in the boreal forest, Atmos. Chem. Phys., 18, 13 231–13 243, https://doi.org/10.5194/acp-18-13231-2018, 2018

Zhang, Y., Wang, Y., Gray, B.A. Gu, D., Mauldin, L., Cantrell, C., Bandy, A., Surface and free tropospheric sources of methanesulfonic acid over the tropical Pacific Ocean, Geophysical Research Letters, 41, 5239-5245, 2014